# RETHINKING SCALE: HOW MULTI-AGENT COLLABORATION ENABLES SMALLER MODELS TO RIVAL GPT-4 IN VIDEO UNDERSTANDING

## ABSTRACT

The rapid development of large language models (LLMs) has brought new perspectives to the field of video understanding. However, existing methods often rely on large-scale proprietary models, such as GPT-4, to achieve competitive performance. This paper challenges the notion that scale is the primary driver of capability by introducing RIVAL, a framework demonstrating how multi-agent collaboration enables smaller open-source models (72B or fewer) to rival their large-scale counterparts. RIVAL consists of two key components: a Multi-stage React Planner (MSRP) for structured stepwise reasoning and Multi-agent Debate Refinement (MADR) for collaborative answer generation. MSRP enhances instruction-following through precise control, while MADR improves answer quality via multi-perspective debate. Using a 72B model, our framework sets a new state-of-the-art on the EgoSchema subset with 66.8% accuracy, surpassing prior GPT-4 based methods by 6.6%. Furthermore, we demonstrate that even smaller open-source models (0.6B to 32B) across the Qwen 2.5 and 3 series achieve competitive performance with RIVAL. We also demonstrate competitive performance on the Next QA benchmark. Highlighting its efficiency, RIVAL can process over 28 hours of continuous video input using limited computational resources.

## 1 INTRODUCTION

With the rapid advancement of multimedia technologies, video understanding has emerged as one of the key tasks in computer vision and has garnered significant attention (Gong et al., 2025; Madan et al., 2024). The rapid development of large language models (LLMs) has brought new perspectives to the field of video understanding. For example, some studies have attempted to transform video understanding tasks into text reasoning problems, leveraging the powerful semantic relationship modeling capabilities of LLMs to infer connections between video frames (Wang et al., 2024c; Ataallah et al., 2024). However, the reliance of existing LLM-based methods on proprietary models introduces two major challenges: data privacy concerns and prohibitive resource requirements.

**Data privacy concerns**: VideoAgent (Wang et al., 2024c), for instance, utilizes LLMs to perform observation and reflection processes to gather information and answer specific queries. However, this single-agent architecture heavily relies on the reasoning capabilities of LLMs (Figure 1 Self-Evaluation). If the generated results deviate, it can directly lead to erroneous answers. To mitigate this issue, VideoAgent relies on large-scale commercial models such as GPT-4. However, in practical applications, exposing user data to external commercial models poses significant privacy risks.
**Resource**: LLoVi (Zhang et al., 2024) converts video segments into textual inputs for LLMs and achieves significant performance improvements by leveraging the strong language reasoning ability. LLoVi requires the model's text window to accommodate the full textual description of the video. When video lengths extend to tens of minutes or even hours, this requirement becomes constrained by the context window size, posing a major bottleneck for understanding long videos (Figure 1 Summary). These limitations motivate a fundamental question: Are large-scale proprietary models truly necessary to achieve competitive performance? This paper tries to answer this question. Specifically, we conduct experiments using small-scale open-source models within a resource-constrained setting, notably limiting the context window to 15,000 tokens. Operating under these constraints, we propose RIVAL, a novel framework that achieves competitive performance with a model as small as 32B parameters.

The core architecture of RIVAL comprises two modules: Multi-stage React Planner (MSRP) and Multi-agent Debate Refinement (MADR). MSRP addresses the weaker reasoning and instruction-following abilities of small-scale models. Specifically, it first decomposes original complex task into simpler sub-tasks. Then, through multi-stage prompting, it guides model's state to transition through these sub-tasks according to pre-defined rules. Notably, within sub-task, we empower the model with retrieval tool, enabling it to fetch keyframes without processing the full video content. MADR subsequently mitigates error propagation from the multi-step reasoning process. Following MSRP, an adversarial debate is initiated in which affirmative and negative agents seek evidence from opposing perspectives to challenge and progressively refine the current answer.

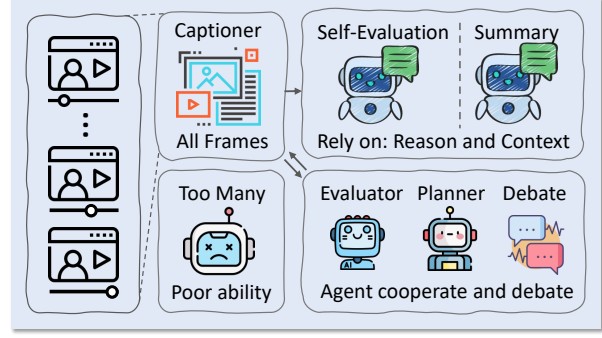

Figure 1: **Illustration of our RIVAL and prior methods.** Top: Prior methods use a single agent with large proprietary models for self-evaluation and summary, posing significant resource and privacy risks. Bottom: RIVAL leverages a cooperative multi-agent system to decompose tasks and achieve competitive performance on small models.

We evaluate RIVAL's performance against prior GPT-4-based methods on the EgoSchema (Mangalam et al., 2023) and Next QA (Xiao et al., 2021) benchmark, deploying the entire Qwen 2.5 and Qwen 3 model series. Our findings show that, despite relying on small-scale models and limited resources, RIVAL delivers competitive and often superior results. Notably, on a subset of EgoSchema, RIVAL with 72B/32B models surpasses the previous state-of-the-art by a substantial margin, achieving 66.8% and 65.0% accuracy—outperforming the GPT-4 baseline by 6.6% and 4.8%, respectively. In addition, RIVAL proves its practical scalability by handling a continuous video input exceeding 28 hours, all while operating on limited resources. The primary contributions are summarized as follows:

- We propose RIVAL, a novel video understanding framework designed to run on small-scale models in resource-constrained environments, without relying on proprietary models.
- The Multi-stage React Planner (MSRP) compensates for the weaker instruction-following of small models. MSRP decomposes complex tasks into manageable sub-tasks and enables tool use for efficient keyframe retrieval, obviating the need for full video processing.
- The Multi-agent Debate Refinement (MADR) mitigates error propagation in multi-step reasoning. This module initiates an adversarial debate between affirmative and negative agents to challenge, verify, and progressively refine answers based on evidence.
- Extensive experiments on the Qwen 2.5 and 3 series demonstrate RIVAL's superiority and practical value. The framework not only sets a new state-of-the-art on benchmarks but also proves its robustness by handling a 28-hour video, showcasing it as a high-performing, resource-efficient, and privacy-preserving solution.

## 2 RELATED WORK

**Long-form Video Understanding.** The field of video understanding has progressed from handcrafted feature-based methods to approaches using deep learning and large language models (LLMs). Early methods relied on features like Improved Dense Trajectories (IDT) (Shu et al., 2015) and Histogram of Oriented Gradients (HOG) (Dalal & Triggs, 2005), processed by models such as Support Vector Machines (SVM) (Hearst et al., 1998). Deep learning introduced architectures like CNNs (Krizhevsky et al., 2012), LSTMs (Hochreiter & Schmidhuber, 1997), and Transformers (Vaswani et al., 2017), with notable advances including TimeSformer (Bertasius et al., 2021) and self-supervised methods like VideoMAE (Tong et al., 2022). Inspired by natural language processing, LLMs are applied to video understanding in three ways: converting videos to text for summarization (Zhao et al., 2024; Chen et al., 2023a; Xue et al., 2024); mapping video frames to text space with instruction embeddings (Shu et al., 2023; Chen et al., 2023b; Ko et al., 2023); and combining textual descriptions with video embeddings (Lin et al., 2023; Han et al., 2023; Wang et al., 2024a). Among these, the first approach is

commonly used for long videos, as it alleviates visual token density issues within LLM text window constraints. However, existing methods often rely on large proprietary models, posing data privacy risks and high computational costs. To address these issues, we propose the RIVAL framework, which uses open-source, lightweight models instead of commercial LLMs.

**Large Language Models Agent.** Large Language Models (LLMs) are widely used for their strong contextual understanding and reasoning abilities (Meng et al., 2025; Wang et al., 2025; Liu et al., 2025). Unlike predefined, rule-driven methods (Huang et al., 2022; Dasgupta et al., 2023), agent-based systems with LLMs offer greater flexibility, enabling dynamic adaptation to complex and evolving scenarios. These agents can interact with various environments, such as games (Valmeekam et al., 2022; Yao et al., 2023b), robotics (Shridhar et al., 2021; Fan et al., 2022), and web applications (Yao et al., 2023a; Trivedi et al., 2024), efficiently integrating information for reasoning and decision-making. Agent-based approaches have demonstrated success in fields like translation (Cui et al., 2025; Chun et al., 2025), medical diagnostics (Sviridov et al., 2025; Maharana et al., 2025), and financial analysis (Lopez-Lira, 2025; Zhu et al., 2025). However, when applied to video understanding, existing methods face challenges such as dependency on extended context windows (Zhang et al., 2024). To address these issues, we propose RIVAL. RIVAL maintains stable reasoning performance and achieves performance comparable to or even surpassing proprietary large-scale commercial models.

**Multi-Agent Debate.** Multi-Agent Collaboration (MAC) is gradually emerging as a critical research direction in the field of natural language processing (NLP) (Chen et al., 2024; Singhal et al., 2023; Wang et al., 2024d). By facilitating information exchange, collaboration, and competition among agents, MAC effectively addresses the limitations of individual agents in reasoning capabilities, thereby significantly enhancing the overall performance and reliability of the system. Within the framework of MAC, Multi-Agent Debate (MAD) has emerged as a distinctive mode of collaboration and has garnered increasing attention (Chan et al., 2024; Smit et al., 2024; Liang et al., 2024). This approach emulates the human debating process, enabling each agent to dynamically present arguments, refute opposing viewpoints, and iteratively refine their own positions. MAD not only broadens the depth and scope of reasoning but also improves the diversity and logical consistency of generated content. Answer generation in ours RIVAL is modeled as a MAD process, where agents are empowered to autonomously retrieve relevant information to substantiate their respective arguments, thereby providing stronger evidential support. Through the debate process, the agents contribute perspectives and analyses, enhancing the comprehensiveness and objectivity of the generated responses.

## 3 METHOD

Our pipeline is illustrated in Figure 2. Consistent with prior work, we decode the video into frames using a fixed sampling rate. Retrieval tools are then used to generate initial information. This information is fed into an iterative process to dynamically add missing details and filter out irrelevant content. During this process, MSRP analyzes the evaluation results and generates usage plans for the retrieval tools. Finally, the updated information is used to produce an initial answer. MADR initiates a debate based on the initial answer, offering multi-perspective insights to refine the initial response.

### 3.1 INFORMATION INITIALIZATION

Since the video content remains almost consistent within short time intervals, we follow prior work (Zhang et al., 2024) and sample the video at a fixed frame rate to obtain a sequence of frames. VideoAgent randomly samples frames at fixed intervals. However, this approach may lead to the selection of irrelevant frames, which could impact subsequent reasoning. To address this, we employ CLIP for image-text alignment to sample frames that are most relevant to the given query:

$$I_s = I_q \cup I_a, \ I_q = \text{Top}_k(\text{Sim}(I, Q)), \ I_a = \text{Top}_k(\text{Sim}(I, A)). \tag{1}$$

Here, I represents the images decoded from the video. Q and A denote the question and the option, respectively. Sim denotes the image-text similarity score, and $\text{Top}_k$ refers to selecting the top $k$ frames with the highest similarity. To extract image information, we use an image description model to generate textual descriptions. This retrieve-then-describe approach reduces the need to process all frames while ensuring an effective starting point for the information retrieval process.

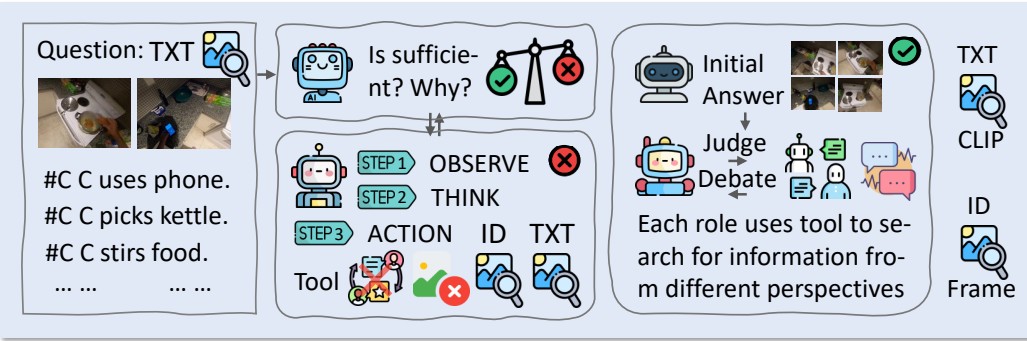

Figure 2: **The pipeline of our RIVAL**. The raw video is decoded into frames, and retrieval tools are used to extract initial information (left). MSRP interacts with evaluators to dynamically add missing information and filter out irrelevant content (middle). Finally, MADR generates an initial answer based on the collected information and initiates a debate process to refine the answer further (right).

## 3.2 MULTI-STAGE REACT PLANNER

In the information initialization phase, we retrieved some information relevant to the problem. However, two main issues remain: (1) the information generated lacks clear distinctions between the characteristics and trade-offs of candidate options, increasing uncertainty in decision-making, and (2) it fails to effectively filter out noise (e.g., irrelevant or low-quality content). To address these issues, we designed a dual-agent collaborative mechanism consisting of an Evaluator and a Planner (Figure 2 middle). This mechanism aims to extract key information and improve decision-making accuracy.

**Evaluator.** The Evaluator is responsible for assessing the collected information on a scale of 1 to 10, where 10 represents the highest quality. It evaluates two aspects—clarity and alignment with problem-related information—weighted at 60% and 40%, respectively. Furthermore, the Evaluator provides explanatory feedback alongside its scores to better guide the Planner in refining the process:

$$\text{Evaluation}(I_s) = \text{LLM}(\text{Prompt}_e(M_c(I_s), \text{Accuracy}, \text{Completeness})), \tag{2}$$

where $M_c$ denotes the description model, which is a large model that generates textual descriptions corresponding to input images. $\text{Prompt}_e$ indicates the pre-defined evaluation instructions tailored to guide the model's output. $\text{Accuracy}$ and $\text{Completeness}$ represent the evaluation criteria, with weights assigned at 60% and 40%, respectively, reflecting their relative importance in the assessment process. The result $\text{Evaluation}(I_s)$ consists of two elements: score, which quantifies the evaluation results numerically, and reason, which provides a justification or explanation for the assigned score.

**Planner (MSRP).** The Planner's main goal is to search for relevant information and remove noise data, providing clearer and more complete input for the task. In our framework, the Planner works with the Evaluator to create plans that maximize the Evaluator's score, ensuring alignment with task requirements. However, current LLMs show limitations when using the ReAct (Yao et al., 2023c), unlike commercial-grade models. For example, they may skip deep reasoning when making plans or stop at reasoning without taking actions, which reduces task efficiency. To address these challenges, we propose a Multi-Stage ReAct Planner (MSRP). MSRP decomposes the reasoning and action phases of the ReAct into multiple predefined sub-stages, forcing the model to progress step-by-step through the stages while performing explicit state transitions. This approach effectively mitigates inconsistencies between the reasoning and action processes and enhances the quality of plan:

$$\text{State}_{\text{THINK}} = \text{LLM}(\text{Prompt}_p(\text{State}_{\text{init}}, \text{Evaluation}(I_s), \text{OBSERVE})). \tag{3}$$

In the $\text{Prompt}_p$, we instruct the LLM to generate plans in three distinct phases: OBSERVE, THINK, and ACT. The initial state, denoted as $\text{State}_{\text{init}}$, represents the starting condition for generating the OBSERVE phase. Since OBSERVE is the first phase, $\text{State}_{\text{init}}$ is initialized as empty. OBSERVE specifies the task to be performed in the current state. We can replace $\text{State}_{\text{init}}$ with $\text{State}_{\text{THINK}}$ and assign the task of THINK to generate $\text{State}_{\text{ACT}}$. In the final state, we provide the LLM with a list of callable tools and instruct it to generate a usage plan based on the available tools:

$$\text{Plan} = \text{LLM}(\text{Prompt}_p(\text{State}_{\text{ACT}}, \text{Evaluation}(I_s), \text{ACT}, \text{Tool}_p)), \tag{4}$$

where we define four tools ($\text{Tool}_{\text{p}}$): (1) Stop Searching, which terminates the search loop prematurely instead of waiting until the maximum number of searches is reached; (2) Delete by Frame ID, which removes irrelevant information to enhance the clarity of the collected data; (3) Add by Frame ID, which extracts images corresponding to the specified frame ID, generates descriptions using $\text{M}_{\text{c}}$, and adds them; and (4) Add by Text, which queries CLIP to match the most similar frame based on the input text, generates descriptions using $\text{M}_{\text{c}}$, and adds them. By generating a structured tool-usage plan, MSRP can refine the task-related information, enhancing the accuracy of subsequent answers. Subsequently, the plan is executed through function calls to modify the content of the information.

This process of evaluation, planning, and execution is repeated iteratively until a predefined condition is satisfied. Specifically, we define three conditions for termination: (1) the Planner calls the Stop Searching method for the loop, (2) the score assigned by the Evaluator exceeds a predefined threshold, or (3) the maximum number of iteration steps is reached. The loop ends when any condition is met:

$$\text{Cond}_{\text{lp}} = (\text{Score}_{\text{e}} > \alpha) \vee (\text{Call Stop Searching}) \vee (\text{Loop Number} > \text{Max Number}). \quad (5)$$

### 3.3 Multi-agent Debate Refinement

The Evaluator collaborates with the Planner to generate clear and precise descriptions of the problem. We utilize this information to address the given question. As shown in Figure 2 (right), we first employ an agent to generate an initial answer. However, compared to the commercial model, the initial answer exhibits significant shortcomings in logic and consistency. Furthermore, while our approach of using step-by-step state transitions for reasoning is highly efficient, it increases the risk of error accumulation. To address these issues, we propose the Multi-Agent Debate Refinement (MADR) to refine and optimize the initial answer. Through MADR, Agents with different roles analyze the initial answer from multiple perspectives, incorporating diverse viewpoints to supplement and correct it. This multi-perspective refinement process directly corrects logical fallacies, resolves inconsistencies, and mitigates the accumulation of errors from the initial reasoning stages.

**Initial Answer.** In a straightforward manner, we consolidate all previously collected information into the Agent's prompt, enabling it to generate an answer. Similar to the Evaluator, we require the Agent to provide the reason behind its answer when generating the response:

$$\text{Initial Answer} = \text{LLM}(\text{Prompt}_{\text{IA}}(\text{M}_{\text{c}}(\text{I}_{\text{IA}}))), \quad (6)$$

where $\text{Prompt}_{\text{IA}}$ represents the instruction provided to the LLM to given the answer. $\text{I}_{\text{IA}}$ denotes the frame IDs obtained during the prior information retrieval phase.

**Responder (MADR).** Given an initial answer, we initiate a debate process to refine it. Our debate process consists of three core roles: the affirmative side, the opposition, and the judge. Within this framework, the affirmative side is responsible for supporting the given answer, while the opposition is tasked with challenging it. When presenting counterarguments, the opposition is required to propose an alternative option. If the opposition prevails, the proposed alternative replaces the initial answer:

$$\text{Statement} = \text{LLM}(\text{Prompt}_{\text{db}}(\text{M}_{\text{c}}(\text{I}_{\text{IA}}), \text{Task}, \text{Statement}_{\text{op}}, \text{Tool}_{\text{db}}), \quad (7)$$

where $\text{Task}$ refers to the role-specific objectives. The affirmative side is responsible for supporting the given answer, while the opposition is tasked with opposing it. $\text{Statement}_{\text{op}}$ represents the statement generated by the opposing role. In $\text{Prompt}_{\text{db}}$, we require both the affirmative side and opposition to consider the opposing arguments when generating their statements. Additionally, they are prompted to utilize the tool $\text{Tool}_{\text{db}}$ to search for evidence supporting their respective positions.

We allow both the affirmative side and the opposition to call the tool only once per response. The list of available tools contains two options: (1) Frame ID Query Tool. This tool retrieves the description corresponding to a given frame ID by extracting the image associated with the ID and then generating a description using $\text{M}_{\text{c}}$. (2) Text Query Tool. This tool uses CLIP to retrieve the frame ID corresponding to a given textual query and then generates a description using $\text{M}_{\text{c}}$. Finally, the judge evaluates the statements presented by both the affirmative side and the opposition during each round of the debate. The debate terminates when the judge determines that either a consensus has been reached or one side has prevailed. Otherwise, the debate continues until the pre-defined maximum number of rounds is reached. Thus, the conditions for terminating the debate are as follows:

$$\text{Cond}_{\text{db}} = \text{Agreement} \vee \text{Win} \vee (\text{Number Round} > \text{Max Number}), \quad (8)$$

Table 1: **Comparison with other state-of-the-art methods on the EgoSchema benchmark.** We provide two sets of comparisons. The left side presents the performance of our RIVAL method compared to other approaches based on LLM/vLLM, while the right side outlines comparisons with other training-based or large-scale proprietary models. #parm indicates the number of parameters.

| Method | Model | Subset | Full | Method | Subset | Full |
|---|---|---|---|---|---|---|
| MoReVQA (Min et al., 2024) | PaLM 2 | - | 51.7 | Random Chance | 20.0 | 20.0 |
| ProViQ (Choudhury et al., 2023) | GPT 3.5 | 57.1 | - | Bard + ImageViT | 35.0 | 35.0 |
| IG-VLM (Kim et al., 2024) | GPT 4V | 59.8 | - | + ShortViViT | 42.0 | 36.2 |
| MVU (Ranasinghe et al., 2024) | LLaVA 13 B | - | - | + PALI (Papalampidi et al., 2024) | 44.8 | 39.2 |
| ViViT (Papalampidi et al., 2024) | ViViT | 56.8 | 33.3 | FrozenBiLM (Yang et al., 2022) | - | 26.9 |
| SeViLA (Yu et al., 2023) | BLIP-2 | 25.7 | 22.7 | InternVideo (Wang et al., 2022) | - | 32.1 |
| Vamos (Wang et al., 2024b) | GPT 4 | 51.2 | 48.3 | GPT 4 Turbo | 31.0 | 30.8 |
| LLoVi (Zhang et al., 2024) | GPT 4 | 57.6 | 50.3 | GPT 4V (Wang et al., 2024c) | 63.5 | 55.6 |
| VideoAgent (Wang et al., 2024c) | GPT 4 | 60.2 | 54.1 | Gemini 1.0 Pro (Team et al., 2024) | - | 55.7 |
| **Ours:** RIVAL (72B) | Qwen 2.5 | 66.8 | 56.4 | **Ours:** RIVAL (Qwen 2.5) | **66.8** | 56.4 |
| **Ours:** RIVAL (32B) | Qwen 3 | 65.0 | **57.2** | **Ours:** RIVAL (Qwen 3) | 65.0 | **57.2** |

where Agreement and Win represent two judgment outcomes made by the judge: Agreement indicates that both sides have reached a consensus, while Win signifies that one side has prevailed over the other. Number Round refers to the number of rounds in the debate. If the debate ends without reaching a consensus, meaning the maximum number of rounds has been reached, the judge must choose a winner from either the affirmative side or the opposition.

The pseudocode for MSRP and MADR is provided in Appendix A (Algorithm 1 and 2).

## 4 EXPERIMENTS

### 4.1 DETAILS

Consistent with prior work Zhang et al. (2024); Wang et al. (2024c), we sample video frames at 1-second intervals. For experiments conducted on EgoSchema, we utilize LaViLa Zhao et al. (2023) as the video description model. To prevent data leakage, we employ a version of the model trained on Ego4D with all segments overlapping with EgoSchema removed. For Next-QA, we leverage CogAgent as the description model, aligning with previous studies Zhang et al. (2024); Wang et al. (2024c). As the retrieval model, we adopt EVA-CLIP-8B plus Sun et al. (2023), using the frame with the highest cosine similarity as the matched item during the retrieval process. For the LLM, we utilize the open-source Qwen-2.5/-3 series for all experiments. To ensure compatibility with OpenAI's standard workflow, we utilize vllm Kwon et al. (2023) to deploy the LLM on an A100 GPU (80GB) and configure the maximum number of tokens for inference to 15,000. Furthermore, due to the inability of the 72B model to be deployed on a single A100 GPU, we leverage vllm to enable tensor parallelism and perform inference across two A100 GPUs (totaling 160GB). All weights of Qwen2.5 are obtained from the official repository of the ModelScope community. For all model scales, we utilize instruction-tuned versions, such as Qwen2.5-72B-instruct.

### 4.2 MAIN RESULT

**EgoSchema.** In Table 1, we present the performance comparison between our RIVAL method and other SOTA method on the EgoSchema benchmark. Compared to smaller-scale methods such as MoReVQA, ProViQ, and IG-VLM, our RIVAL demonstrates significant performance advantages. On the subset, RIVAL achieves an accuracy improvement of 10 points over ViViT and 31.1 points over SeViLA. Similarly, RIVAL maintains its superiority with an 11-point accuracy margin ahead of MVU. When compared to approaches based on GPT 4, our method also showcases competitive advantages. On the subset, RIVAL outperforms Vamos by 15.6 points, LLoVi by 9.2 points, and VideoAgent by 6.6 points. On the full dataset, RIVAL achieves consistent superiority with an accuracy margin of 8.1, 6.1, and 2.3 points over Vamos, LLoVi, and VideoAgent, respectively. Even when compared to proprietary models, RIVAL holds a leading position. On the subset and full dataset, RIVAL surpasses

Table 2: **Comparison with other state-of-the-art (SOTA) methods on the Next QA benchmark.** The ATP-Hard subset represents the validation set, which is a more challenging subset. The upper portion of the table presents methods based on supervised training, while the lower portion lists methods utilizing LLM/vLLM. Our RIVAL demonstrates significant performance advantages.

| Method | #parm | Validation Set | | | | ATP-hard subset | | |
|---|---|---|---|---|---|---|---|---|
| | | All | Causual | Temporal | Descriptive | All | Causual | Temporal |
| Random Chance | - | 20 | 20 | 20 | 20 | 20 | 20 | 20 |
| VFC (Yang et al., 2021) | 164 M | 52.3 | 49.6 | 51.5 | 63.2 | - | - | - |
| ATP (Buch et al., 2022) | 88 M | 54.3 | 53.1 | 50.2 | 66.8 | 38.8 | 38.4 | 36.5 |
| MIST (Gao et al., 2023b) | 88 M | 57.2 | 54.6 | 56.6 | 66.9 | - | - | - |
| GF (Bai et al., 2024) | 88 M | 58.8 | 56.9 | 57.1 | 70.5 | 49.3 | 48.7 | 50.3 |
| CoVGT (Xiao et al., 2023) | 149 M | 60.7 | 59.7 | 58.0 | 69.9 | - | - | - |
| SeViT (Kim et al., 2023) | 215 M | 56.7 | 54.0 | 54.1 | 71.3 | - | 43.3 | 46.5 |
| HiTeA (Ye et al., 2023) | 297 M | 63.1 | 62.4 | 58.3 | 75.6 | - | 47.8 | 48.6 |
| VFC (Yang et al., 2021) | 540 B | 51.5 | 51.6 | 45.4 | 64.1 | 31.4 | 32.2 | 30.0 |
| InternVideo (Wang et al., 2022) | - | 49.1 | 43.4 | 48.0 | 65.1 | - | - | - |
| AssistGPT (Gao et al., 2023a) | 1.8 T | 58.4 | 60.0 | 51.4 | 67.3 | - | - | - |
| ViperGPT (Surís et al., 2023) | 175B | 60.0 | | | | - | - | - |
| SeViLA (Yu et al., 2023) | 4 B | 63.6 | 61.3 | 61.5 | 75.6 | - | - | - |
| LLoVi (Zhang et al., 2024) | 1.8 T | 67.7 | 69.5 | 61.0 | 75.6 | - | - | - |
| VideoAgent (Wang et al., 2024c) | 1.8 T | 71.3 | 72.7 | 64.5 | 81.1 | 58.4 | 57.8 | 58.8 |
| **Ours:**RIVAL (Qwen 2.5) | 72 B | **74.4** | **76.3** | **67.5** | 82.4 | **66.5** | **70.2** | **61.2** |
| **Ours:**RIVAL (Qwen 3) | 32 B | 73.2 | 74.1 | 67.0 | **82.9** | 63.7 | 66.2 | 60.1 |

GPT 4V by 3.3 and 0.8 points, respectively. Furthermore, RIVAL is training-free, yet it achieves significant performance gains over other training-based approaches. Specifically, RIVAL outperforms FrozenBiLM (Yang et al., 2022) by 25.5 points and InterVideo (Wang et al., 2022) by 23.3 points.

**Next-QA.** Table 2 presents the performance comparison between our RIVAL method and other SOTA approaches on the Next-QA benchmark. Compared to supervised training-based methods, RIVAL demonstrates significant advantages. Specifically, when compared to methods using the ViT-B-32 (Dosovitskiy et al., 2021) backbone, including ATP (Buch et al., 2022), MIST (Gao et al., 2023b), and GF (Bai et al., 2024), our RIVAL achieves performance improvements of 20.1, 17.3, and 15.6 points respectively. On the more challenging ATP-Hard subset, the performance gap further widens, with RIVAL outperforming ATP and GF by 27.7 and 17.2 points, respectively, highlighting its superior capability in difficult scenarios. When compared to methods based on LLM/vLLM, RIVAL also achieves considerable performance margins, outperforming VFC and SeViLA (Yu et al., 2023) by 22.9 and 10.8 points, respectively. Even in comparison with approaches utilizing GPT 4, our RIVAL maintains its leading position. On the validation set, RIVAL achieves accuracy improvements of 18.3 points over AssistGPT (Gao et al., 2023a), 7 points over LLoVi, and 3.1 points over VideoAgent. Furthermore, on the challenging subset, RIVAL achieves even more substantial performance differences, surpassing VFC and VideoAgent by 35.1 and 8.1 points, respectively.

## 4.3 COMPARISON ACROSS VARIOUS SCALES

To further evaluate the practical performance of our RIVAL, we present in Table 3 a performance comparison across various scales. Notably, to assess performance on long videos, we concatenated all videos from the EgoSchema subset to form a single video approximately 28 hours in length. Based on this long video, we evaluated the performance of answering questions from the EgoSchema subset, which is labeled as Long in Table 3. Furthermore, we implemented VideoAgent using the full Qwen 2.5 model series (see Appendix C). First, RIVAL demonstrates remarkable capital efficiency. Across both the Qwen 2.5 and 3 series, it consistently matches the performance of prior methods like LLoVi and VideoAgent but with significantly smaller models. For instance, on the Qwen 3 series, a 1.7B RIVAL model can achieve performance comparable to much larger counterparts. Second, in direct comparisons on identical models, RIVAL consistently and significantly outperforms VideoAgent. This performance gap widens on more challenging tasks. On the EgoSchema evaluation dataset, for

Table 3: **Performance comparison across various scales of LLM.** Scale indicates the parameter size of the LLM, all of which are from the Qwen 2.5/3 series. Subset refers to the validation set of EgoSchema. All videos in EgoSchema were concatenated into a single approximately **28-hour-long** video to evaluate the performance on all questions in the Subset, Long (28h). Additionally, we provide implementation of VideoAgent using the Qwen 2.5 series in Appendix C.

| Scale | EgoSchema | | Next QA (Val) | | | | Next QA (ATP-Hard) | | |
|---|---|---|---|---|---|---|---|---|---|
| | Long (28h) | Subset | All | Causual | Temporal | Descriptive | All | Causual | Temporal |
| Qwen 2.5 Series: 14 B ≈ LLoVi; 32 B ≈ VideoAgent | | | | | | | | | |
| 72 B | 48.7 | 66.8 | 74.4 | 76.3 | 67.5 | 82.4 | 66.5 | 70.2 | 61.2 |
| 32 B | 44.8 | 61.4 | 72.4 | 74.4 | 65.1 | 80.4 | 63.9 | 68.0 | 58.0 |
| 14 B | 45.2 | 57.6 | 70.5 | 72.4 | 63.3 | 78.4 | 62.2 | 65.9 | 56.8 |
| 7 B | 41.4 | 53.2 | 66.9 | 68.2 | 59.9 | 76.2 | 58.1 | 61.4 | 53.3 |
| 3 B | 38.9 | 53.0 | 59.0 | 59.2 | 54.8 | 67.1 | 50.3 | 53.0 | 46.6 |
| 1.5 B | 33.8 | 43.6 | 53.6 | 55.2 | 49.8 | 56.6 | 45.9 | 49.3 | 41.1 |
| Qwen 3 Series: 1.7 B ≈ LLoVi; 8 B ≈ VideoAgent | | | | | | | | | |
| 32 B | 46.0 | 65.0 | 73.2 | 74.1 | 67.0 | 82.9 | 63.7 | 66.2 | 60.1 |
| 14 B | 46.2 | 60.4 | 71.9 | 72.7 | 66.3 | 80.7 | 62.7 | 65.3 | 58.9 |
| 8 B | 47.1 | 60.4 | 70.8 | 72.4 | 63.9 | 79.5 | 61.5 | 64.3 | 57.5 |
| 4 B | 45.2 | 59.2 | 68.8 | 69.5 | 63.0 | 78.3 | 59.7 | 61.9 | 56.7 |
| 1.7 B | 41.3 | 56.7 | 62.9 | 63.0 | 57.7 | 57.7 | 52.7 | 54.5 | 50.1 |
| 0.6 B | 35.7 | 45.6 | 53.9 | 53.3 | 49.8 | 64.5 | 45.1 | 47.3 | 41.8 |

example, RIVAL achieves up to a 10-point accuracy improvement over VideoAgent (using Qwen 2.5 72B). Finally, RIVAL shows superior robustness in extreme-length video scenarios. On the 28-hour concatenated video, VideoAgent's performance collapses to near-random levels. In stark contrast, RIVAL maintains robust performance with only minor degradation, outperforming VideoAgent by as much as 14.8 points and proving its capability for real-world, long-form video analysis.

## 4.4 CASE STUDY

In Figure 3, we present a case study comparing RIVAL and VideoAgent. VideoAgent follows a response-reflection process. However, when the initial response contains errors, the subsequent reflection process is prone to failure. As shown in the middle section of Figure 3, the initial response provides an incorrect answer. During the subsequent reflection step, LLM assigns a confidence score of 3 to the answer, indicating that it fails to recognize the mistake in its generated response. Consequently, the final answer reproduces the same error. In contrast, our RIVAL framework successfully identifies and corrects this error. After the initial information retrieval phase, MSRP conducts further searches to gather additional information relevant to the given question. However, similar to VideoAgent, RIVAL also generates an incorrect initial response. This error is later identified and resolved during the subsequent MADR process. Specifically, the counterarguments generated during the debate process successfully identify the inconsistency. The counterarguments highlight that, based on the observed information, no details link entity C's behavior to Option 1, and they provide a more plausible alternative answer. The judge module, considering the counterarguments, determines that the counterarguments are more persuasive, revising the final answer accordingly.

## 5 CONCLUSION

This paper proposed RIVAL, a novel framework that demonstrates how smaller, open-source models can achieve competitive and often state-of-the-art performance in video understanding by addressing the key limitations of current methods. Through its two components—MSRP for structured, tool-assisted reasoning and MADR for adversarial answer refinement—RIVAL operates effectively as a training-free solution. Crucially, RIVAL circumvents the primary bottlenecks for practical application: its ability to process video without full-frame ingestion mitigates context window limitations, while its independence from proprietary APIs addresses significant data privacy concerns. This allows

Question: After completing the painting task, what actions does c take to prepare for their next activities on the construction site?
0. C cleans up tools, washes hands, uses phone. 1. C cleans up tools, dries hands, uses phone. 2. C cleans up tools, sanitizes hands, makes a call.
3. C cleans up tools, applies lotion, uses phone. 4. C cleans up the tools, washes his hands, and uses his phone.          Answer: 0

Given a video that has 180 frames, the frames are decoded at 1 fps. Given the following descriptions of five uniformly sampled frames in the video:
{'frame 1': '#C C brushes the metal mesh ', 'frame 45': '#C C paints metal fence', 'frame 90': '#C C paints the iron mesh with a paint brush', 'frame 135': '#O man G walks in', 'frame 180': '#C C touches the phone'}, ...,
# C to denote the sentence is an action done by the camera wearer (the person who recorded the video while wearing a camera on their head). #O to denote that the sentence is an action done by someone other than the camera wearer. Please answer the question.
-----Response----
Best Answer: Choice 1 is the most comprehensive and logical, as it includes careful tool storage, thorough hand washing, and preparing for the next task by putting on the phone.\n\nTherefore, the best answer is :{"final_answer": "1"}

Please assess the confidence level in the decision-making process. The provided information is as as follows. Given a video that has 180 frames, the frames are decoded at 1 fps. Given the following descriptions of five uniformly sampled frames in the video: {Captions}, {'Answere': "1"}. Evaluate based on the relevance, completeness, and clarity of the provided information in relation to the decision-making context.
-----Response----
{"final_answer": "1", "confidence": "3"}.  ⊗

Retrieve Frame Based on Question and Options:
[{'segment_id': '135', 'frame': '134 - 135', 'description': '#O man X stands beside a wall'}, {'segment_id': '50', 'frame': '49 - 50', 'description': '#C C scrapes paint from the metal bar to the grill with the paint scraper in his right hand.'}, ...]
*****Evaluator*****
You are an AI assistant tasked with evaluating the Accuracy (60%) and Completeness (40%) of provided descriptions answering a given question. You must assign scores strictly between 1-10 (10 = best).
-----Response----
{"score": 6, 'Resson': "The actions focus on ending the task but miss cleaning and preparation steps, making them incomplete."

*****Planner (MSRP)*****
You are an AI assistant. Follow the Observe-Think-Act methodology to generate actionable plans. At each step, ensure your reasoning and actions align with maximizing task outcomes.
STEP 1: OBSERVE
Carefully observe the provided input data, including the video description list, evaluation results, and previous plan results. {captions}...
-----Response----
Statement_OBSERVE :Most of the actions involve C performing various tasks related to painting and cleaning. The current descriptions are somewhat incomplete and lack details about critical post-painting activities.
STEP 2: THINK
Based on the observed facts from the previous step, generate actionable strategies for solving the task. Statement_OBSERVE
-----Response----
Statement_THINK : Enhance Description Completeness. Reasoning: The current descriptions lack details about post-painting activities, particularly cleaning or sanitizing hands.

*****Planner (MSRP)*****
STEP 3: ACT
Implement strategies developed in the previous phase by making function calls that optimize the task outcome. Statement_THINK
-----Response----
Plan: add_frame_base_id: [160, 165, 170]
add_frame_base_on_sentences:
["C thoroughly washes his hands.", "C applies hand sanitizer."]

*****Initial Answer******
Given a video that has 180 frames, the frames are decoded at 1 fps. Given the following descriptions of the sampled frames in the video. Please think carefully and write the best answer index in Json format: {'answer': 'xxx', 'reason': 'xxx'}. {updated captions}
-----Response----
{'answer': '1', 'reason': 'it aligns with necessity of thorough handwashing, adhering to best practices for task preparation.'}.

***** Responder (MADR *****
***Affirmative Side ***
You are a debater representing the affirmative side, defending given answer. {updated captions}. you have one chance to call function.
-----Tool----
Function call result: query_frame_base_on_sentences:
['C puts the paint can down', 'C removes the paint brush from the tin', 'C picks the phone from the fence with his right hand']
-----Response----
{"answer": "1", "evidence": "The provided frames show C putting the paint can down, removing the paint brush from the tin, and picking up the phone. These actions align with carefully putting away Tools and preparing for the next activity."}
***Opposing Side ***
You are a highly analytical debater representing the opposing side. {updated captions}. you have one chance to call function.
-----Response----
{"answer": 0, "evidence": "The provided descriptions do not mention C washing, drying, sanitizing, or lotions his hands. Only actions related to putting away the paint brush and tin, and picking up the phone are described."}
*** Judge ***
You are an judge tasked with determining most accurate answer based on arguments. {history} {updated captions}.
-----Response----
{final_answer: 0, "reason": "The Negative side's evidence clearly states that the provided descriptions only mention C putting the paint can down, removing the paint brush from the tin, and picking up the phone. There is no evidence of C washing, drying, sanitizing, or applying lotion to his hands. Therefore, the most accurate answer based on the given evidence is option 0."}  ✓

Figure 3: **Case Study of VideoAgent and RIVAL on EgoSchema Benchmark (Qwen 2.5 72B).** The top part displays the given question and corresponding options. In the middle section, the reasoning process of VideoAgent is presented, which highlights its step-by-step analysis approach. Despite including a self-assessment process, it fails to identify its own errors. The bottom section illustrates our RIVAL framework. RIVAL first retrieves information and then initiates a debate process based on the retrieved information, generating opinions from diverse perspectives to refine the answer.

RIVAL to robustly handle videos of arbitrary and even extreme length, paving the way for more accessible, efficient, secure, and truly scalable video understanding in real-world scenarios.

## 6 REPRODUCIBILITY STATEMENT

Due to our company's data security policies, we are currently unable to release the full source code. Upon completion of the internal security review, we plan to make the corresponding experimental logs and replication scripts publicly available. In the interim, to facilitate reproducibility, we provide detailed pseudocode for both MSRP and MADR in the Appendix. Furthermore, all prompt templates used in our study are detailed in the subsequent sections. For dataset configurations and content descriptions, we refer readers to the official open-source repositories of VideoAgent and LLoVi. We believe these resources provide a sufficient basis for replicating our key findings.

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

# A PSEUDOCODE

---

**Algorithm 1:** Frame Sampling and Multi-Stage React Planner (MSRP)

---

/* Step 1: Parameters and Initialization */
  $V$: Input video, $Q$, $A$: Question/answer, $k$: Top-k selector, $\text{Tool}_p$: Toolset
  Max_Number: Maximum iterations, $\alpha$: Score threshold
/* Step 2: Frame Sampling */
  $I_s$: Relevant frames set
  $I \leftarrow \text{DecodeFrames}(V)$ // Extract frames from video
  $I_q \leftarrow \text{Top}_k(\text{Sim}(I, Q))$ // Select top-k frames for question
  $I_a \leftarrow \text{Top}_k(\text{Sim}(I, A))$ // Select top-k frames for answer
  $I_s = I_q \cup I_a$ // Combine frames relevant to question and answer
  $\text{GenerateDescriptions}(I_s)$ // Generate descriptions using model $M_c$
/* Step 3: Multi-Stage Planning */
  $\text{State}_{init} = \emptyset$ // Initialize planning state
  **while** $\neg(\text{Score}_e > \alpha \vee \text{Stopped} \vee \text{Loop} > \text{Max\_Number})$ **do**
      // Observing Phase //
      $\text{State}_{\text{OBSERVE}} \leftarrow \text{LLM}(\text{Prompt}_p(\text{State}_{init}, I_s, \text{"OBSERVE"}))$
      // Thinking Phase //
      $\text{State}_{\text{THINK}} \leftarrow \text{LLM}(\text{Prompt}_p(\text{State}_{\text{OBSERVE}}, I_s, \text{"THINK"}))$
      // Acting Phase //
      $\text{Plan} \leftarrow \text{LLM}(\text{Prompt}_p(\text{State}_{\text{THINK}}, I_s, \text{"ACT"}, \text{Tool}_p))$
      $\text{ExecutePlan}(\text{Plan}, \text{Tool}_p, M_c)$
      $\text{Score}_e, \text{Reason} \leftarrow \text{Evaluation}(I_s)$
      $\text{State}_{init} \leftarrow \text{State}_{\text{THINK}}$
  **end**
// * Evaluation Function * //
  $\text{Evaluation}(I_s)$:
    $\text{Scores} \leftarrow \text{AssessClarity}(I_s, 60\%) + \text{AssessAlignment}(I_s, 40\%)$
    **return** $\text{Scores}, \text{Feedback}$ // Evaluation result
// * Final Output * //
  **return** $I_{\text{IA}}$

---

**Algorithm 2:** Multi-Agent Debate Refinement (MADR)

---

/* Step 1: Initial Answer Generation */
$\text{Answer}_{initial} \leftarrow \text{LLM}(\text{Prompt}_{\text{IA}}(M_c(I_{\text{IA}}), \text{Question}, \text{Options}))$ // Generate answer with reasoning
/* Step 2: Multi-Agent Debate */
  Define roles: Affirmative, Opposition, Judge
  $\text{Statement} \leftarrow \text{Answer}_{initial}$ // Initialize debate input
**while** $\text{Cond}_{db}$ **do**
    $\text{Affirmative Statement} \leftarrow \text{LLM}(\text{Prompt}_{db}(\text{Task}_{\text{Affirmative}}, \text{Statement}, \text{Tool}_{db}))$
    // Support answer //
    $\text{Opposition Statement} \leftarrow \text{LLM}(\text{Prompt}_{db}(\text{Task}_{\text{Opposition}}, \text{Statement}, \text{Tool}_{db}))$
    // Challenge answer //
    $\text{Decision} \leftarrow \text{Judge}(\text{Assess}(\text{Affirmative Statement}, \text{Opposition Statement}))$
    // Judge evaluates arguments //
    $\text{Round Number} \leftarrow \text{Round Number} + 1$ Increment round
**end**
// * Debate Termination Conditions * //
$\text{Cond}_{db} \leftarrow \text{Agreement} \vee \text{Win} \vee (\text{Round Number} > \text{Max Number})$
  Agreement: Both sides reach consensus
  Win: One side prevails over the other
  Round Number: Tracks debate iterations
// * Final Output * //
  **return** $\text{Answer}_{updated}$

---

Table 4: **Performance comparison across various scales of LLM (VideoAgent).** Scale indicates the parameter size of the LLM, all of which are from the Qwen 2.5 series. Subset refers to the validation set of EgoSchema. All videos in EgoSchema were concatenated into a single approximately **28-hour-long** video to evaluate the performance on all questions in the Subset, Long (28h).

| Scale | EgoSchema | | Next QA (Val) | | | | Next QA (ATP-Hard) | | |
|-------|-----------|--------|------|---------|----------|-------------|------|---------|----------|
|       | Long (28h) | Subset | All  | Causual | Temporal | Descriptive | All  | Causual | Temporal |
| 72 B  | 33.8 | 56.8 | 71.0 | 72.1 | 63.8 | 82.6 | 62.3 | 65.7 | 57.5 |
| 32 B  | 33.0 | 51.4 | 69.0 | 70.5 | 61.1 | 80.2 | 60.0 | 64.2 | 53.9 |
| 14 B  | 33.2 | 51.2 | 69.6 | 71.1 | 62.3 | 78.8 | 61.5 | 65.9 | 54.9 |
| 7 B   | 31.6 | 47.8 | 66.4 | 67.0 | 60.7 | 75.3 | 58.2 | 61.1 | 53.7 |
| 3 B   | 29.4 | 48.0 | 55.0 | 54.6 | 49.8 | 66.4 | 46.4 | 48.7 | 43.2 |
| 1.5 B | 23.4 | 32.8 | 49.1 | 47.7 | 47.7 | 61.6 | 42.1 | 42.8 | 40.8 |

## B  DATASETS AND METRICS

**EgoSchema** Mangalam et al. (2023). EgoSchema is a standard benchmark designed for long-video understanding. It consists of approximately 5,000 real-world video samples, covering various human activities such as cooking, crafting, and sports. Each video is unedited, captured from a first-person perspective, and has a minimum duration of 3 minutes. The dataset's total duration exceeds 250 hours, with an average video length of 3 minutes and 17 seconds. For each video, a multiple-choice question with five candidate options is provided. EgoSchema is specifically designed to evaluate long-video understanding in zero-shot settings. The dataset is organized into two parts: a subset and the full set. The subset contains 500 QA tasks with corresponding annotations, while the full set includes all video. Performance on the full set requires submitting results to the official server.

**Next QA** Xiao et al. (2021). Next QA is a video understanding benchmark designed to evaluate a model's capabilities in causal reasoning, temporal analysis, and scene understanding. The dataset consists of 5,440 videos, with an average duration of 44 seconds. The content of the videos includes scenes of daily activities and interpersonal interactions. A total of 52,000 question-answer (QA) pairs are annotated for these videos, with causal reasoning, temporal analysis, and scene understanding accounting for 48%, 29%, and 23% of the QA tasks, respectively. Consistent with prior work, we conduct our analysis only on a subset of Next QA. In addition, we further partitioned this subset into a more challenging ATP-hard subset Buch et al. (2022). QA tasks in the ATP-hard subset cannot be resolved using single-frame images and are designed to assess long-term reasoning ability.

## C  VIDEOAGENT PERFORMANCE BASED ON QWEN

In a direct comparison against VideoAgent on the same Qwen series models, RIVAL shows a significant performance advantage (Figure 4). For instance, on Qwen 2.5 72B, we achieved a performance improvement of 3.4. Across other scales, RIVAL maintains a consistent lead. Even on Qwen 2.5 1.5B, RIVAL still outperforms VideoAgent by 4.5 in accuracy. On more challenging benchmarks, such as EgoSchema's evaluation dataset, the performance gap between RIVAL and VideoAgent further widens. For example, on Qwen 2.5 72B, RIVAL achieved a 10-point improvement in accuracy. In the ultra-long video evaluation scenario (Long), our method significantly surpassed VideoAgent in performance. When facing nearly 28 hours of video input, VideoAgent's performance noticeably deteriorates; for Qwen 2.5 1.5B, its accuracy drops to 23.4, barely above random selection. In contrast, our RIVAL manages to interpret the 28-hour video with only minor performance degradation. On Qwen 2.5 1.5B, RIVAL still achieved an accuracy of 33.8, outperforming VideoAgent by nearly 10%.

## D  ABLATION STUDY

Parameter Ablation Study for RIVAL. In Figure 4, we present the parameter ablation experiments for RIVAL. Five hyperparameters are analyzed, including the Top-k value for the initial search, the threshold for terminating loops in MSRP, the maximum number of iterations, as well as the threshold and number of steps for initiating the debate process. It is important to highlight that

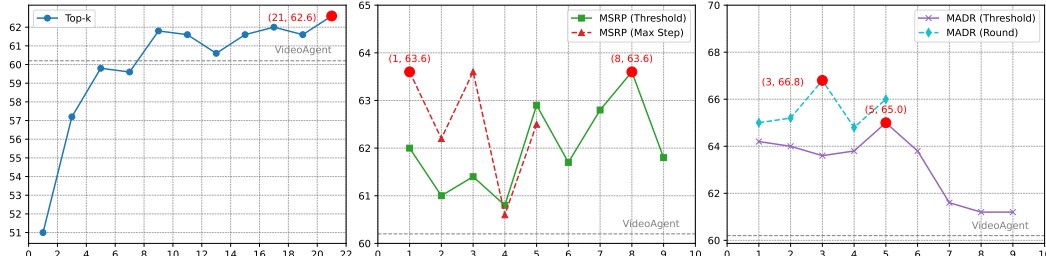

Figure 4: **Parameter Ablation Experiments on EgoSchema Benchmark.** From left to right, the figures depict the parameters for initial information retrieval, the threshold for terminating loops in MSRP, and the maximum number of iterations. The final figure illustrates the threshold for answer refinement and the maximum number of debate rounds. In each figure, the best performance achieved by the SOTA GPT-4-based VideoAgent on the EgoSchema is indicated by gray dashed lines.

RIVAL demonstrates robustness and effectiveness. After completing the initial information search, the remaining parameter settings are designed to explore the upper bounds of RIVAL's performance. As shown in the second and third subfigures, RIVAL is capable of outperforming the previous state-of-the-art GPT-4-based VideoAgent even under relatively unfavorable parameter settings.

**Top-k Value Analysis.** The impact of the top-k parameter on model performance is illustrated in the first subfigure of Figure 4. During the initial information collection stage, both the question and answer are used to retrieve corresponding frames, which are subsequently described using a Caption model. Intuitively, retrieving more frames provides additional information, offering greater support for subsequent answer generation. However, excessive information may lead to redundancy, negatively affecting the model's decision-making. As shown in the first subfigure of Figure 4, initially increasing the top-k value leads to a significant improvement in model performance, with accuracy rising sharply from 51% at top-k = 1 to nearly 62% at top-k = 9. Beyond this point, while performance continues to improve, it becomes slower and exhibits slight fluctuations until top-k reaches 21. Therefore, to avoid introducing excessive irrelevant information and optimize performance, we select top-k = 21 as our final setting, at which the model achieves an accuracy of 62.6%.

**Threshold for MSRP Loops.** The second subfigure of Figure 4 illustrates the impact of the loop threshold in MSRP, represented by the green solid line. After the initial information is retrieved, the planner and evaluator collaborate to gather additional information while filtering out irrelevant data. The loop threshold determines whether the iterative process should continue. Intuitively, applying the loop to all available information is expected to yield better results, which aligns with our experimental findings. As shown in Figure 4, overall model performance improves as the loop threshold increases. These results further validate the rationale behind the collaborative loop between the planner and evaluator. Based on this observation, we select a loop threshold of 8.

**Iteration Steps in MSRP Loops.** In the second subfigure of Figure 4, we use a red dashed line to illustrate the impact of the number of iteration steps on model performance. A greater number of iteration steps results in longer contextual content. However, since we use the 72B network as the baseline model, its ability to support extended contexts is limited compared to larger models, such as GPT 4. Consequently, model performance gradually decreases as the number of iterations increases. To mitigate issues arising from excessively long contextual content, we set maximum number to 1.

**Threshold for Debate.** The third subfigure of Figure 4 shows the trend of model performance as the debate threshold varies, represented by the purple solid line. During the debate process, we observed that applying debate to all questions with different scores had varying impacts on performance. Specifically, debating low-scoring questions was less effective compared to setting a higher threshold and only debating high-scoring ones. For instance, using a threshold of 1, where answers with evaluator scores greater than 1 are debated, was found to be less effective than setting the threshold to 5. Through visualization, we discovered that low-scoring information often fails to perfectly capture the correct corresponding options, making it prone to being dominated by the opposing side during the debate process. Therefore, we set the threshold to 5, ensuring that only results with scores greater than 5 are fine-tuned through debate.

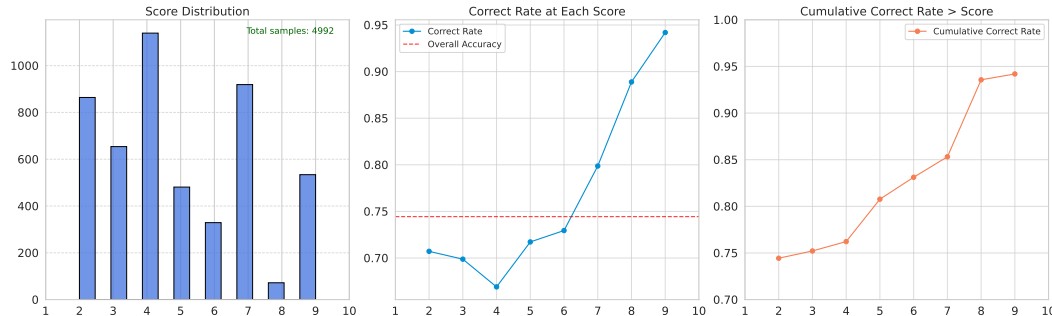

Figure 5: **Score Analysis on the Next QA Benchmark.** From left to right, the figure presents the distribution of scores after iterative processing, the accuracy corresponding to each score value, and the accuracy for scores greater than the current score. For example, the accuracy within the range (2, 10]. Overall, the accuracy of RIVAL improves as the evaluation score increases.

**Rounds of Debate.** The third subfigure of Figure 4 illustrates the trend of model performance as the number of debate rounds increases, represented by the dashed line. The performance of RIVAL improves with an increasing number of debate rounds. When the number of rounds is set to 3, the performance reaches its peak value of 66.8, which aligns with the results we reported in the experimental section. Therefore, we hypothesize that increasing the number of debate rounds could further enhance the performance of RIVAL, as long as the model's ability to comprehend contextual information remains uncompromised. For simplicity, we set the number of debate rounds to 3.

## E    EVALUATION ANALYSIS

Whether guiding decision-making for the planner in MSRP loops or using thresholds in the MADR model to filter low-scoring samples, the effectiveness of our method, RIVAL, is fundamentally dependent on the accuracy of the evaluation scores. To validate this dependence, we conducted a series of data analyses on scores generated by the evaluator, as shown in Figure 5. Due to the limited sample size of the EgoSchema benchmark, which contains only 500 samples, the representativeness of the data is constrained, leading to larger deviations in score distribution. To further ensure the reliability of our analysis, we extended the experiments to the Next QA benchmark, which includes nearly 5,000 samples and offers broader representativeness.

The analysis of this dataset reveals that a larger proportion of questions are clustered in the lower score ranges, where accuracy is typically lower. Specifically, when the evaluation score equals 2, the prediction accuracy of RIVAL is approximately 71%, whereas at a score of 9, the accuracy rises to nearly 95%. This finding indicates a strong positive correlation between the evaluator's scoring capabilities and the accuracy of RIVAL predictions. From an interval-based perspective, this trend is particularly prominent: in lower score intervals (e.g., a score of 2), the accuracy is around 75%, while in higher score intervals (e.g., a score of 9), the accuracy quickly climbs to nearly 95%.

The collaborative workflow of RIVAL, which integrates the evaluator and planner, proves to be both reasonable and efficient. By decomposing the overall system capability into multiple independent but mutually cooperative components, we are able to reduce the context length while leveraging the advantages of multi-agent collaboration, thereby unlocking the potential of collective intelligence.

## F    PROMPT DESIGN

Prompt Design in Evaluator (Figure 6). Figure 6 illustrates the prompt design in the Evaluator. In the prompt, we specify the evaluation criteria for the LLM and assign weights of 60% and 40%, respectively. For each scoring level, evaluation standards are explicitly defined. Furthermore, good and bad examples are provided to users within the prompt. The output format is restricted to JSON, and the model is required to include explanations for its outputs.

You are an AI assistant tasked with evaluating the accuracy and completeness of provided descriptions answering a given question. Focus on identifying the correctness, relevance, and clarity of the descriptions, as well as whether they have enough detail to fully answer the question. You must assign scores strictly between 1-10 (10 = best).

#### **Scoring Guidelines**
1. **Accuracy (60% weight):** How well the description matches the question.
   - **1-2:** Irrelevant, incorrect, or misleading.
   - **3-4:** Weak alignment, unclear or partially wrong.
   - **5-6:** Moderately accurate but incomplete or noisy.
   - **7-9:** Mostly accurate with minor imperfections.
   - **10:** Perfectly correct and directly answers the question.

2. **Completeness (40% weight):** Does the description provide enough detail to fully answer the question?
   - **1-2:** Extremely vague or lacking critical details.
   - **3-4:** Missing important aspects, limited information.
   - **5-6:** Partially complete, with notable omissions.
   - **7-9:** Mostly complete, minor missing details.
   - **10:** Fully detailed with all relevant context included.

### **Your Output Format**
Respond strictly in JSON format.

System Prompt

Task Context:
 - Description List Format (For Reference):
 A list containing segment dictionaries with:
   - "segment_id": string identifier (e.g., "1")
   - "frame": video time range (e.g., "0:05-0:10")
   - "description": text with #C/#O tags

 - Action Tags:
 #C = Camera wearer's action\n#O = Others' actions

 - Required Output Format:
 JSON object with:
   - "score": number between 1-10
   - "reason": scoring rationale

 - Example of a correct response:
 {
   "score": 8,
   "reason": "The descriptions partially answer the question but lack some key details."
 }

 Input Data:
 - Historical Evaluations: {eval_history}
 - Current Description List: {captions}
 - Question: {Question}
 - Options: {Optoins}
 Requirement:
 1. Evaluate the current description list's accuracy in answering the question using the provided Scoring Guidelines.
 2. Before assigning the final score, ALWAYS perform a comparison between the current description and the following examples:

   **Positive Example:** "#C picks up a red pencil from the table (frame: 0:05-0:10)"
   - This description is clear, detailed, and fully answers the question with subject, object, action, and time frame. set score to 8

   **Negative Example:** "#C does something (frame: 0:05-0:10)"
   - This description is vague, lacks specific details, and fails to directly address the question. set score to 2

 3. Based on this comparison, justify whether the current description aligns closer to the positive or negative example and how this influenced your scoring. Your reasoning in the `"reason"` field MUST include reference to this comparison.

User Prompt

Figure 6: **Prompt Design in Evaluator.** Variable parameters are treated as input variables and are highlighted in the figure for clarity. These parameters include evaluation history, existing descriptive information, and the questions and options of the QA tasks.

Prompt Design in MSRP (Figure 7). Figure 7 presents the prompt design within MSRP. In MSRP, we take the output from the Evaluator as the evaluation direction. This process unfolds in three steps, each corresponding to a distinct stage of the ReAct paradigm. At each stage, the LLM is required to complete the task specified for that stage. For example, in Stage 2, the LLM is prompted to reflect on how to improve the current descriptive information based on prior observations, and this reflection is provided as a strategy for the next stage. In the final stage, the Agent is granted the ability to invoke tools and instructed to implement the previous strategy through function calls. These designs enforce the LLM to execute plan generation tasks strictly in alignment with the ReAct paradigm.

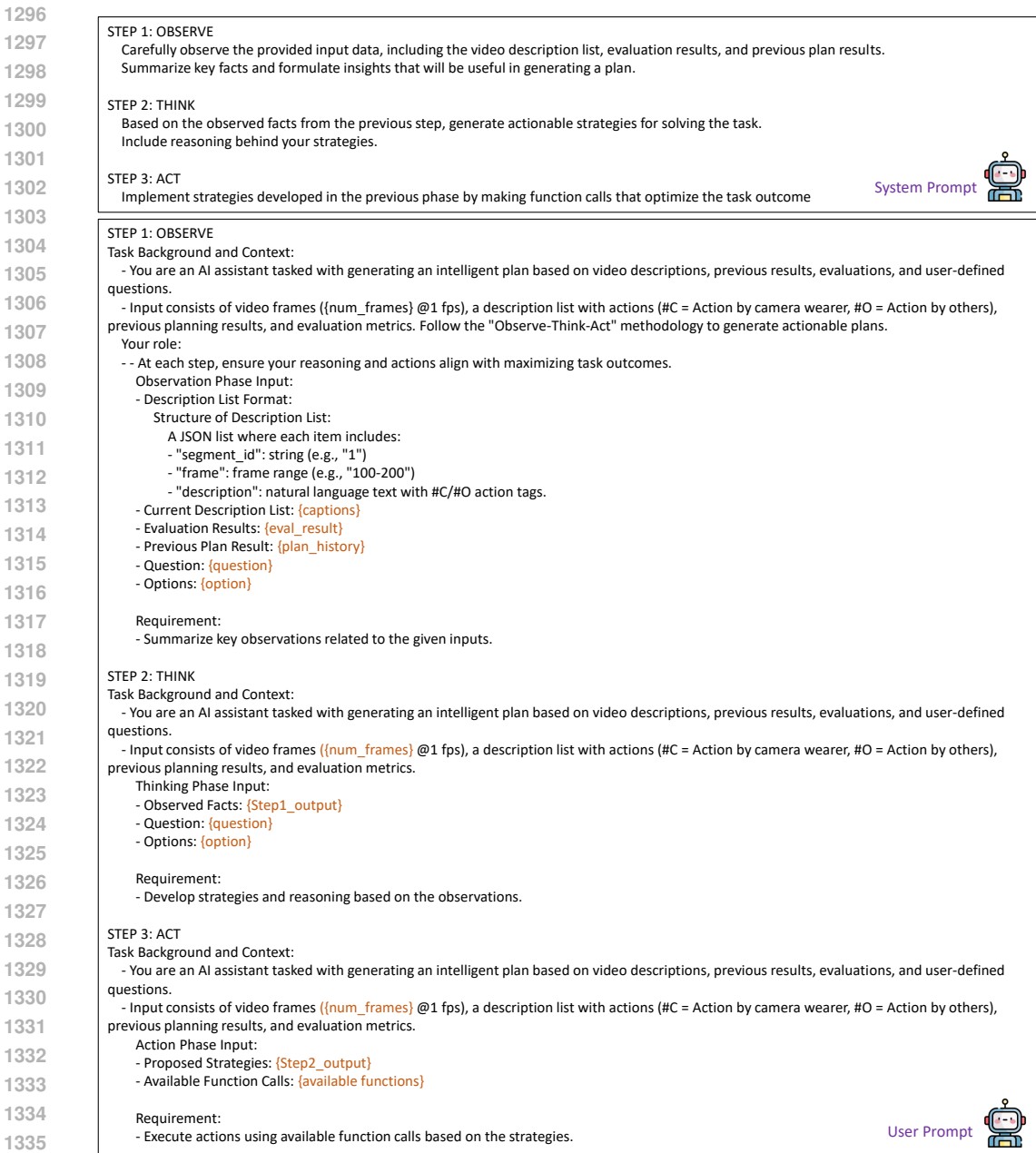

STEP 1: OBSERVE
  Carefully observe the provided input data, including the video description list, evaluation results, and previous plan results.
  Summarize key facts and formulate insights that will be useful in generating a plan.

STEP 2: THINK
  Based on the observed facts from the previous step, generate actionable strategies for solving the task.
  Include reasoning behind your strategies.

STEP 3: ACT
  Implement strategies developed in the previous phase by making function calls that optimize the task outcome                    System Prompt

STEP 1: OBSERVE
Task Background and Context:
  - You are an AI assistant tasked with generating an intelligent plan based on video descriptions, previous results, evaluations, and user-defined questions.
  - Input consists of video frames ({num_frames} @1 fps), a description list with actions (#C = Action by camera wearer, #O = Action by others), previous planning results, and evaluation metrics. Follow the "Observe-Think-Act" methodology to generate actionable plans.
  Your role:
  - - At each step, ensure your reasoning and actions align with maximizing task outcomes.
    Observation Phase Input:
    - Description List Format:
      Structure of Description List:
        A JSON list where each item includes:
          - "segment_id": string (e.g., "1")
          - "frame": frame range (e.g., "100-200")
          - "description": natural language text with #C/#O action tags.
    - Current Description List: {captions}
    - Evaluation Results: {eval_result}
    - Previous Plan Result: {plan_history}
    - Question: {question}
    - Options: {option}

    Requirement:
    - Summarize key observations related to the given inputs.

STEP 2: THINK
Task Background and Context:
  - You are an AI assistant tasked with generating an intelligent plan based on video descriptions, previous results, evaluations, and user-defined questions.
  - Input consists of video frames ({num_frames} @1 fps), a description list with actions (#C = Action by camera wearer, #O = Action by others), previous planning results, and evaluation metrics.
    Thinking Phase Input:
    - Observed Facts: {Step1_output}
    - Question: {question}
    - Options: {option}

    Requirement:
    - Develop strategies and reasoning based on the observations.

STEP 3: ACT
Task Background and Context:
  - You are an AI assistant tasked with generating an intelligent plan based on video descriptions, previous results, evaluations, and user-defined questions.
  - Input consists of video frames ({num_frames} @1 fps), a description list with actions (#C = Action by camera wearer, #O = Action by others), previous planning results, and evaluation metrics.
    Action Phase Input:
    - Proposed Strategies: {Step2_output}
    - Available Function Calls: {available functions}

    Requirement:
    - Execute actions using available function calls based on the strategies.                    User Prompt

Figure 7: **Prompt Design in MSRP.** In the MSRP framework, variable parameters are treated as input variables and highlighted in the figure for clarity. These variable parameters include evaluation history, existing descriptive information, and the questions and options associated with QA tasks.

Prompt Design for Initial Answer Generation and Debate Roles of Proponents and Opponents (Figure 8). For simplicity, during the initial answer generation process, we use prompts provided by the VideoAgent that include only basic background information, such as the number of video frames. The LLM generates responses directly while being required to provide reasoning for its predictions. In the debate prompt design, we employ a similar structure for both proponents and opponents. Their prompts explicitly outline their respective tasks: the proponent searches for information to support the provided answer, while the opponent seeks information to refute it. Furthermore, both sides are granted the ability to use tools. During each turn of the debate, they are allowed to invoke a tool once. To streamline the process, both sides utilize the same user template.

System Prompt
You are a helpful assistant designed to output JSON.
User Prompt
Given a video that has {num_frames} frames, the frames are decoded at 1 fps. Given the following descriptions of the sampled frames in the video:
  {caption}
  #C to denote the sentence is an action done by the camera wearer (the person who recorded the video while wearing a camera on their head).
  #O to denote that the sentence is an action done by someone other than the camera wearer.
  Please answer the following question:
  ```
  {question}
  ```
  Please think carefully and write the best answer index in Json format {'answer': 'xxx', 'reason': 'xxx'}. Note that only one answer is returned for the question, and ** you must select one answer index from the candidates (0/1/2/3/4)**.

You are a debater representing the affirmative side, defending the given answer. Your task:
  1. Use descriptors or known facts to justify the answer.
  2. Refute negative arguments by identifying flaws and assumptions.
  3. Strengthen your stance with expanded explanations if necessary.
  4. Default to supporting the provided answer unless evidence strongly contradicts it.
  5. For each responce, you have one chance to call the function to collect more information to support you answer.

  Output in JSON:
  {
    'answer': the option you defend,
    'evidence': supporting evidence (≤50 words).
  }

System Prompt

You are a highly analytical debater representing the opposing side. Your task:
  1. Aggressively challenge the given answer by identifying logical flaws, overgeneralizations, contradictions, or gaps in evidence.
  2. Critically analyze even subtle weaknesses in affirmative reasoning.
  3. If proposing an alternative answer, provide compelling evidence and reasoning to justify it.
  4. Avoid speculative or baseless challenges; all objections must be rooted in logic, context, or provided evidence.
  5. When no flaws exist and evidence is sound, reluctantly agree with the affirmative side.
  6. For each responce, you have one chance to call the function to collect more information to support you answer.

  Output in JSON:
  {
    "answer": 0/1/2/3/4, the option you believe is correct (mandatory; no null or none allowed),
    "evidence": Clear and concise supporting evidence, including detailed reasoning behind your objection or
          agreement (≤50 words).
  }

System Prompt

Task Context: a video that has {num_frames} frames, the frames are decoded at 1 fps. Frames are sampled from the video, and their descriptions are extracted: {caption}. #C to denote the sentence is an action done by the camera wearer (the person who recorded the video while wearing a camera on their head). #O to denote that the sentence is an action done by someone other than the camera wearer.

  Question: {question}.

  Given init answer: {init_answer}.

User Prompt

Figure 8: **Prompt Design for the Initial Answer Generation and Roles of Proponents and Opponents in the Debate.** The variable parameters are treated as input variables and are highlighted in the figure for clarity. These parameters include existing descriptive information and the questions and options of the QA tasks. To prevent exceeding the context window, we constrain their evidence to within 50 words.

Prompt Design for the Judge (Figure 9). Figure 9 illustrates the prompt design for the judge. The judge performs two main functions: determining whether the debate process should continue and generating the final answer. For process determination, the judge simply evaluates whether the proponent and opponent have reached a consensus. For generating the final answer, the judge collects the debate's historical information and formulates the final answer accordingly.

# G    Free-From Case

Our RIVAL framework is designed as a universal cognitive architecture capable of transferring to various applications beyond video QA. To verify RIVAL's generalization capability to non-selection tasks, we provide a qualitative case study on a free-form generation task. Prior to utilizing RIVAL for free-form examples (e.g., video summarization or open-ended questions), minimal architectural modifications were required to shift the modules' objectives from selection to generation. As detailed in Table 5, these adjustments primarily involve redefining the information retrieval and final output stages. CLIP Retrieval's objective is switched from distinguishing options to collecting

```
System Prompt (in Debate)
You are an impartial moderator tasked with evaluating a debate between two sides: Affirmative and Negative. Your task:
    1. Objectively assess the strength of reasoning, evidence, and rebuttal effectiveness from both sides, without default bias.
    2. Favor Affirmative if its arguments hold up under scrutiny and Negative fails to provide strong counterarguments.
    3. Favor Negative if its objections are well-reasoned, logical, and backed by compelling evidence that decisively undermines Affirmative's
answer.
    4. If Negative fails to select an answer in the 0-4 range, Affirmative wins by default.
    5. Always provide a clear and neutral summary of your reasoning for the decision.
    Output in JSON:
    {
      "continue": True/False,
      "choice": "Affirmative"/"Negative",
      "reason": "Neutral and concise summary of the decision-making rationale."
    }
User Prompt (in Debate)
Q & A Mission Statement: {}, Affirmative: {} Negative: {}
```

```
System Prompt (Give Answer)
You are an impartial judge tasked with determining the most accurate answer based on arguments presented by both sides. Your task:
    1. Select the option supported by the strongest evidence, reasoning, and overall argument quality.
    2. **Give priority to consensus when both sides recommend the same answer; only evaluate the quality of evidence and arguments when their
answers differ.**
    3. Ensure your judgment is based purely on logical and factual merit, avoiding any bias or assumptions.
    4. Provide a clear and neutral summary of your reasoning for the chosen answer.
    Output in JSON:
    {
      "final_answer": 0/1/2/3/4,
      "reason": "Neutral and concise summary of judgment rationale."
    }
User Prompt (Give Answer)
Based on the debate, select the most accurate answer.
  Output:
  {
    'final_answer': 0/1/2/3/4,
    'reason': 'Reason for final choice.'
  }
```

Figure 9: **Prompt Design for Debate Roles.** The variable parameters are treated as input variables and are highlighted in the figure for clarity. These parameters include the questions, options of the QA tasks, and the dialogue history of both sides in the debate. Note that the decision of whether to continue the debate and the subsequent generation of the final answer are separated processes. However, as both serve summarization functions, we classify them under the same role.

Table 5: Architectural Modifications for Adapting RIVAL from Multiple-Choice Question (MCQ) to Free-form Video Summarization.

| Component | Current Implementation (MCQ Task) | Necessary Modification (Summarization Task) | Rationale |
|---|---|---|---|
| **CLIP Retrieval** | Retrieve frames relevant to *Question* and *Options*. | Retrieve frames relevant to the *Question* and *Chronological Events*. | Ensures collection of key temporal boundaries, not just option-specific details. |
| **MSRP Planner Objective** | Objective: Refine information to clearly distinguish between candidate options. | Objective: Refine information to capture the most **chronologically accurate** and **structurally important events**. | Shifts focus from binary distinction to holistic content coverage. |
| **MADR Agents Debate Focus** | Debate Focus: *Which option is correct?* (Adversarial Selection). | Debate Focus: *Is the generated summary complete, coherent, and accurately supported by the retrieved events?* (Adversarial Verification). | Maintains high-quality control by checking for logical flaws and coherence in the generated text. |
| **Final Judge Output** | Outputs a single choice index ($0/1/2/3/4$). | Outputs a **coherent, flowing narrative** compiled from the validated events and descriptions. | Replaces selection with robust text generation. |

chronologically relevant and event-based information, ensuring the capture of critical temporal boundaries. The MSRP Planner objective shifts from distinguish between candidate options to capture the most chronologically accurate and structurally important events. The Final Judge output is changed from selecting a single index to generating a coherent, flowing narrative compiled from validated events. By redefining these objectives, we successfully converted an EgoSchema QA task

**Question:** Identify the most critical moments in the video and explain how they are essential to the video's overarching narrative.
**Reference Answer:** The most critical moments in the video are when c chops the broccoli, washes the chopping board, and stirs the food

**Retrieve Frame Based on Question:**
[{"segment_id": "36", "frame": "36 - 36", "description": "#C C puts the broccoli in the pot"}, {"segment_id": "101", "frame": "101 - 101", "description": "#C C stirs food in the pan"} {"segment_id": "195", "frame": "195 - 195", "description": "#C C stirs the vegetables with the cooking spoon"}, {"segment_id": "197", "frame": "197 - 197", "description": "#C C stirs the food"},{"segment_id": "199", "frame": "199 - 199", "description": "#C C stirs the vegetables with the cooking spoon, {"segment_id": "172", "frame": "172 - 172", "description": "#C C picks up whisk from the wooden stand"...]

**\*\*\*\*\*Evaluator (Based on Question and Sentences)\*\*\*\*\***
You are an AI assistant tasked with evaluating the Accuracy (60%) and Completeness (40%) of provided descriptions answering a given question. You must assign scores strictly between 1-10 (10 = best).
-----Response----
{"score": 6, "reason": "The score is 6 because the list is moderately accurate but significantly incomplete. The actions chosen are indeed pivotal to understanding the story. However, overall answer is critically flawed because it only provides the what and entirely omits why, making it an incomplete response to the full prompt.}

**\*\*\*\*\*Planner (MSRP, Based on Question) \*\*\*\*\***
You are an AI assistant. Follow the Observe-Think-Act methodology to generate actionable plans. At each step, ensure your reasoning and actions align with maximizing task outcomes.
STEP 1: OBSERVE
Carefully observe the provided input data, including the video description list, evaluation results, and previous plan results. {captions}...
-----Response----
Statement$_{OBSERVE}$ :The descriptions outline a cooking narrative with four phases: preparation, cooking, cleaning, and serving. The list contains redundancies, such as multiple stirring actions, and disconnected events, like the use of measuring tools without clear context.
STEP 2: THINK
Based on the observed facts from the previous step, generate actionable strategies for solving the task. Statement$_{OBSERVE}$
-----Response----
Statement$_{THINK}$ : The plan is to refine this narrative by consolidating repetitive actions, inferring the purpose behind each step (e.g., washing for safety, stirring for technique), and structuring the events to highlight the serving moment as the story's climax.

**\*\*\*\*\*Planner (MSRP)\*\*\*\*\***
STEP 3: ACT
Implement strategies developed in the previous phase by making function calls that optimize the task outcome. Statement$_{THINK}$
-----Response----
Plan: delate_frame_base_id: [193, 195, 199]
add_frame_base_on_sentences:
["C adds broccoli pieces to the cooking pot", "C uses a whisk to mix batter."]

**\*\*\*\*\*\*Initial Answer (free-from text)\*\*\*\*\*\***
Given a video that has 180 frames, the frames are decoded at 1 fps. Given the following descriptions of the sampled frames in the video. Please think carefully and write the best answer index in Json format: {'answer': 'xxx',} {updated captions}
-----Response----
{'answer': The most critical moments in the video depict the full arc of a k itchen cooking narrative. It begins with ingredient preparation, as the cook adds broccoli and other vegetables into pots and pans (frames 36–40). This is followed by hygiene maintenance, including washing and rinsing the chopping board and cleaning cooking utensils such as knives, spoons, and chopsticks.}

**\*\*\*\*\* Responder (MADR, Based on answer text and question) \*\*\*\*\***
**\*\*\*Affirmative Side \*\*\***
You are a debater representing the affirmative side, defending given answer. {updated captions}. you have one chance to call function.
-----Tool----
Function call result: query_frame_base_on_sentences:
[ "C adds broccoli and other vegetables into pots and pans", "C thoroughly washes the chopping board and utensils at the sink", ...]
-----Response----
{"answer": "The most critical moments in the video depict the kitchen cooking narrative from preparation to serving and tool preparation for baking.", "evidence": "Frames show broccoli preparation, utensil cleaning, stirring, serving, and baking tool setup."}

**\*\*\*Opposing Side \*\*\***
You are a highly analytical debater representing the opposing side. {updated captions}. you have one chance to call function.
-----Response----
{"answer": "The truly critical moments are confined to the observable and completed cooking narrative: preparing the vegetables, cooking the meal, serving the final dish, and cleaning the utensils. The so-called 'tool preparation for baking' is a disconnected and unsubstantiated event, not a critical part of the main story.", "evidence": "The 'baking' conclusion is a speculative leap, not an observed fact; tools are merely picked up, not used. Critical moments must contribute to the primary narrative's outcome. "}

**\*\*\* Judge \*\*\***
You are an judge tasked with determining most accurate answer based on arguments. {history} {updated captions}.
-----Response----
{"final_answer": "The most critical moments are the observable and completed actions that form the core cooking narrative: preparing the vegetables, cooking the meal by stirring, serving the final dish, and cleaning the utensils.", "reason": "Since the evidence only shows the tools being picked up, not used, this action lacks the necessary narrative weight. Therefore, the most accurate and evidence-based answer is the one that confines the critical moments to well-supported, four-stage cooking process presented by Negative side."}

Figure 10: **Qualitative Case Study of RIVAL on a Free-form Generation Task.** The visualization demonstrates RIVAL's modularity and generalizability by adapting it from a MCQ task to an open-ended narrative generation task on an EgoSchema sample. The architecture successfully leverages the MSRP Planner to refine evidence and the MADR Debate to verify the chronological consistency, resulting in a coherent, evidence-based final answer.

into a free-form generation task, which requires the model to identify and explain the critical moments in the video narrative. The results of this adaptation are presented in Figure 10.

The findings in Figure 10 demonstrate RIVAL's successful adaptation to the generative task, validating the framework's strong generality. MSRP's Objective Adaptation: The MSRP Planner successfully adapts its THINK and ACT steps to focus on the macro-narrative flow and stages (e.g., post-cooking cleanup) rather than mere option differentiation. This ensures the retrieved information possesses structural importance over time. MADR's Narrative Verification: The MADR shifts its debate focus

from Which option is correct? to Is the generated narrative complete, coherent, and accurately supported by the retrieved events? In the case study, the Opposing Agent successfully challenged a logical flaw in the Initial Answer, pointing out that a certain conclusion was a speculative leap. Free-form Output: The Final Judge module then synthesizes the MADR-validated evidence to produce a logically consistent and evidence-grounded high-quality free-form answer. This successful transference of the RIVAL paradigm from a selection mode to a narrative generation mode confirms RIVAL's modularity and inherent generative capacity. It validates that RIVAL provides small models with not only objective reasoning capabilities but also a flexible form of generative intelligence.

## H  THE USE OF LARGE LANGUAGE MODELS

In this work, we utilized Large Language Models (LLMs) exclusively for polishing the text of the manuscript.

