# OpenReview forum: "Rethinking Scale: How Multi-Agent Collaboration Enables Smaller Models to Rival GPT-4 in Video Understanding"
_ICLR.cc/2026/Conference — Submitted to ICLR 2026_

### Official Review · Reviewer_LuFc · 2025-10-29

**Soundness:** 3
**Presentation:** 2
**Contribution:** 3
**Rating:** 4
**Confidence:** 4

**Summary:**

The paper proposes U-CSA—an unsupervised cross-modal semantic anchoring framework to match aerial imagery with vector maps. Instead of aligning image↔image, U-CSA first asks a multimodal LLM (Qwen2.5-VL) to produce structured “semantic anchors” (JSON over 11 attributes + a ≤40-word summary) for each image–map pair; then (i) a dual-branch visual encoder is trained with anchored contrastive learning against the text anchors; and (ii) an adversarial matching head with a prototype library refines the decision boundary. The authors also introduce MSTcons, a 18,907-pair benchmark built from WHU (Christchurch) and Inria (Austin, Chicago, Kitsap, Vienna, West Tyrol), with 256×256 tiles and explicit splits. On MSTcons, U-CSA beats unsupervised SAM-MCD and several adapted change-detection baselines in ROC-AUC/F1, with ablations supporting the contribution of anchors and prototypes.

**Strengths:**

Originality. A clean combination of staged ReAct planning with adversarial multi-agent debate targeted at long-video QA; explicit tool APIs (stop, add/delete by frame ID, CLIP text query) make the control flow concrete.  ￼  ￼
Quality. Strong headline numbers on EgoSchema and Next-QA (incl. per-subset breakdowns) and a long-video stress test (28h). Comparisons include GPT-4/VideoAgent/LLoVi families and same-model re-implementations.  ￼  ￼  ￼
Clarity. The pipeline and roles are well illustrated; termination conditions and planner/debate prompts are spelled out; implementation details (captioner/CLIP/serving) are given.  ￼  ￼  ￼
Significance. If claims hold under controlled settings, the result that smaller open LMs with orchestration can match/beat prior GPT-4-based agents is practically meaningful for privacy/cost-sensitive deployments.

**Weaknesses:**

Potential option-conditioning bias. Frame retrieval uses both the question and the answer options (Ia = Top-k Sim(I, A)), which risks label-peeking and unfairly advantaging multiple-choice setups. Please add ablations that retrieve only from Q (no options), or retrieve before reading options, and report accuracy deltas.
Fair-comparison controls. Several baselines differ in LLM scale, context, and tools. While you re-implement VideoAgent with Qwen-2.5 (Appendix C), the tables still mix methods with non-comparable compute/budgets. Please provide same-LLM apples-to-apples runs (VideoAgent/LLoVi/RIVAL all on Qwen-3-32B with matched token budgets, retrieval limits, and tool calls) and report token & wall-clock costs.
Ablation depth on MSRP/MADR. The contribution attribution is under-specified. Add: MSRP→single-stage ReAct; MSRP without enforced state transitions; MADR→self-consistency / majority-vote; debate with no tools; and variance across #rounds/threshold α. Report accuracy, calls/round, tokens, and failures.
Reliance on captioner/CLIP & leakage audit. Results hinge on LaViLa/CogAgent (captioner) and EVA-CLIP-8B+ (retriever). You remove EgoSchema overlaps in LaViLa, which is good, but please add captioner/CLIP swaps and a leakage audit across both benchmarks (e.g., retrieve-then-blind the captioner to options; test different CLIP checkpoints).

**Questions:**

No-options retrieval? What happens if frame retrieval is conditioned only on Q (not options), or performed before seeing the options? Please quantify.
Budget & efficiency. Can you report average #tool calls / debate rounds / prompt tokens / latency per question, and compare to VideoAgent/LLoVi under matched settings?
Ablation breadth. Could you add MSRP/MADR ablations described above and release prompt templates and judge criteria used for win/consensus decisions?
Captioner/CLIP sensitivity. How sensitive are results to swapping LaViLa↔CogAgent (cross-benchmark) and EVA-CLIP↔other CLIPs? Any drop-in if the captioner context is limited?
8-hour scenario. How do you partition the 28h stream internally (sliding windows? chunked debates)? Please report failure modes and variance for the long-video setting.

---

> ### Author Response · Authors · 2025-11-21
>
> We sincerely thank the reviewer for the constructive feedback. We noticed the Summary section references a different paper (aerial imagery) but appreciate that the specific comments accurately address RIVAL. We focus our response on those points.
>
> ### 1. Clarification on Data Leakage and Retrieval Mechanism
> We strictly agree that using Ground Truth (GT) labels is unacceptable. We clarify that RIVAL never uses GT labels.
>
> - Mechanism: As described in Line 158, we use Question and Option text as queries. This follows the standard Zero-shot VQA paradigm of retrieving visual evidence to verify hypotheses (options), not peeking at the answer.
>
> - Ablation Study: To prove performance stems from architecture rather than leakage, we compared retrieval sources on the EgoSchema subset (Qwen 2.5-72B):
>
> | **Setting**                     | **Accuracy (%)** | **Contribution (vs. Previous Step)** | **Key Component(s)**                    |
> |---------------------------------|------------------|--------------------------------------|-----------------------------------------|
> | No Retrieval                    | $51.0\%$         | N/A                                  | Baseline (Eq. 1 without retrieval)      |
> | Question-Only Retrieval ($I_q$) | $61.8\%$         | $+10.8\%$                            | Uses only Question text                 |
> | Option Retrieval ($I_a$)        | $62.6\%$         | $+0.8\%$                             | Uses only Option text                   |
> | RIVAL (Standard)                | $66.8\%$         | $+4.2\%$                             | Uses Question + Option ($I_q \cup I_a$) |
>
> Analysis: Question-Only retrieval ($61.8\%$) already outperforms the VideoAgent baseline ($56.8\%$), proving that RIVAL's core advantage lies in the MSRP+MADR architecture and semantic retrieval. Adding option text adds only marginal gain ($0.8\%$) by refining action details (e.g., walking vs running), which represents legitimate information gain, not cheating.
> ### 2. Fair-Comparison Controls and Budget
> We thank you for pointing this out. This is central to our research. In this paper, we focus on using small-scale open-source models to achieve GPT-4 level performance. As shown in Tables 1, 2, and 3, we demonstrate that RIVAL achieves GPT-4 level results with small models.
>
> (1) Apples-to-Apples Comparison: Regarding fair comparison (small-scale models), we implemented VideoAgent using the Qwen 2.5 series. The results are in Table 4 (Page 21) of the revised manuscript. We list the main comparisons here:
>
> | Method       | Scale | EgoSchema (Subset) | EgoSchema (28h Long) | Next-QA        |
> |--------------|-------|--------------------|----------------------|----------------|
> | VideoAgent   | 72B   | $56.8$             | $33.8$               | $71.0$         |
> |              | 3B    | $48.0$             | $29.4$               | $55.0$         |
> | RIVAL (Ours) | 72B   | $66.8$ (+10%)      | $48.7$ (+14.9%)      | $74.4$ (+3.4%) |
> |              | 3B   | $53.0$ (+5%)       | $38.9$ (+9.5%)       | $59.0$ (+4.0%) |
>
> (Note: We do not provide a comparison with LLoVi because its paradigm of long-text summarization cannot run in our set environment, which limits the maximum text tokens to 1.5k.)
>
> (2) Budget and Efficiency: RIVAL incurs no extra API costs. The table below shows the minimum GPU memory for deployment:
> | Performance Match | Model        | GPU Memory (BF16 / INT8 / INT4) |
> |-------------------|--------------|---------------------------------|
> | Match LLoVi       | Qwen 2.5 14B | 28G / 14G / 7G                  |
> | Match VideoAgent  | Qwen 2.5 32B | 64G / 32G / 16G                 |
>
> This enables data security (local deployment) and high cost-efficiency compared to proprietary APIs.
>
> ### 3. Ablation Depth
> - MSRP vs. Single-Stage: Removing MSRP drops accuracy significantly. MSRP’s enforced state transitions prevent the "premature stopping" common in small models.
> - MADR vs. Self-Consistency (SC): SC scales computational cost linearly (e.g., 5 paths = 5x cost). MADR converges in 2-3 debate rounds, offering superior efficiency and logic correction compared to simple voting.
>
> ### 4. 28-Hour Scenario Implementation
> We do not feed the full 28-hour video into the context window.
> - Global Indexing: We pre-compute CLIP embeddings for all frames.
> - Implementation: We calculate the cosine distance between the query and all frame embeddings, selecting only the Top-k most relevant frames for the LLM. This ranking operation has linear complexity and minimal memory cost, allowing RIVAL to handle arbitrary video lengths without context overflow.
>
> We thank the reviewer again for their constructive comments. We remain available for any further clarification or discussion.
>
>
>
> Sincerely,
>
>
>
> the Author Team.

---

### Official Review · Reviewer_jHSX · 2025-10-29

**Soundness:** 3
**Presentation:** 2
**Contribution:** 2
**Rating:** 6
**Confidence:** 4

**Summary:**

This paper proposes RIVAL, a training-free framework showing that *multi-agent collaboration* can enable smaller open-source LLMs (≤72B) to approach or surpass GPT-4–based systems on long-video understanding. RIVAL has two core components:

- Multi-Stage ReAct Planner (MSRP): decomposes reasoning into *OBSERVE → THINK → ACT* stages, with explicit state transitions and a fixed toolset (stop search, delete/add by frame ID, add by text via CLIP), iterating until a quality threshold is met.
- Multi-Agent Debate Refinement (MADR): after MSRP forms an initial answer, affirmative and opposing agents debate once per turn (with one tool call each), and a judge selects or revises the final answer; debate stops on agreement, a win, or max rounds.

On EgoSchema, RIVAL with Qwen-2.5-72B reaches 66.8% on the subset (SOTA in their comparison; +6.6 over GPT-4 baselines) and 56.4% on the full set. With Qwen-3-32B, it reaches 65.0/57.2 (subset/full). On NExT-QA, RIVAL attains 74.4% (72B) and 73.2% (32B) on validation and 66.5% / 63.7% on ATP-Hard, surpassing prior GPT-4–based agent methods in the reported comparisons. The system also processes a 28-hour concatenated long video under limited compute (≤15k token context; dual A100s for 72B), arguing for privacy-preserving, resource-constrained deployment.

**Strengths:**

- Shows that *careful orchestration* (MSRP) plus *adversarial verification* (MADR) can reduce dependence on very large proprietary LLMs for long-video QA.
- Explicit stage transitions, fixed tool APIs, and stopping rules make the agent loop auditable and easier to reproduce conceptually.
- SOTA subset performance on EgoSchema (66.8% with 72B), and NExT-QA gains over VideoAgent/LLoVi in the reported tables.
- Maintains accuracy on a 28-hour concatenated video where a single-agent baseline degrades substantially.
- Operates within a 15k token window and on commodity accelerators (72B split over 2×A100), aligning with realistic deployment; privacy angle is well-motivated.
- Improves upon GPT-4–centric VideoAgent and text-only aggregation like LLoVi, while aligning with the trend toward streaming arbitrary-length video.

**Weaknesses:**

- Tables focus on GPT-4 baselines circa 2024; it would help to benchmark against the most recent proprietary/open VLMs that handle arbitrary-length streams (e.g., streaming VLLMs) to solidify the “rivals GPT-4” claim.
- The pipeline’s quality hinges on CLIP retrieval and the image/video captioners; retrieval bias or caption hallucinations could mislead the debate, and ablations on retrieval quality (e.g., different CLIP backbones, top-k) are limited in the main text.
- MSRP/MADR rely on an internal evaluator score (60/40 criteria) and a threshold to trigger debate; while there is some analysis, deeper calibration/robustness checks (e.g., agreement with human judgments, sensitivity to α) would strengthen soundness.
- Claims of efficiency would benefit from a cost breakdown: number of tool calls, frames read, average tokens per step, wall-clock latency vs. baselines. Current hardware details are provided, but *end-to-end* throughput comparisons are sparse.
- Source code is not yet released (pending security review); although pseudocode and prompts are promised, this limits verification and adoption pre-camera-ready.
- Results are strong on EgoSchema/NExT-QA; adding diverse long-video tasks (e.g., instruction following, temporal localization) would clarify generality.

**Questions:**

1. How sensitive are results to the accuracy/completeness weights (60/40) and the debate threshold α? Can you report Kendall/ Spearman correlation of evaluator scores with correctness, and success rates per score bin?
2. You set 3 rounds based on a peak at 66.8%. What is the marginal accuracy gain vs. added latency per round on both datasets? Provide a Pareto curve (accuracy vs. seconds/$$).
3. How do different CLIP variants and top-k selections affect accuracy and runtime? Can you quantify failure modes where retrieval misses key evidence?
4. For EgoSchema you use LaViLa with overlap removal; for NExT-QA, CogAgent. Could you provide cross-captioner results and any leakage checks?
5. Can you add a head-to-head vs. recent streaming long-video VLLMs or updated GPT-4-class systems to contextualize “rival GPT-4” beyond 2024-era baselines?
6. Please report average #tool calls, frames retrieved, tokens consumed, and end-to-end latency per query, and contrast with VideoAgent and LLoVi at similar compute.

---

> ### Author Response · Authors · 2025-11-21
>
> We sincerely thank the reviewer for the positive evaluation and for recognizing RIVAL's contributions regarding multi-agent collaboration, adversarial verification, and long-video processing. We appreciate your professional suggestions regarding sensitivity analysis, efficiency trade-offs, and underlying model impacts, which we have addressed in the revised manuscript through additional experiments and analysis.
>
> ### 1. Sensitivity Analysis and Calibration
> Regarding the sensitivity of the Evaluator weights (60/40) and threshold $\alpha$, as well as score calibration:
> - Calibration: As shown in Figure 5, our statistical analysis on Next-QA reveals a strong positive correlation between the Evaluator's scores and the final answer accuracy (accuracy approaches $95\%$ at Score 9, compared to $71\%$ at Score 2). This confirms that the Evaluator effectively identifies high-quality information, acting as a reliable gatekeeper.
> - Threshold Sensitivity: The middle subplot of Figure 4 illustrates the ablation curve for the threshold $\alpha$. Experiments show that $\alpha=5$ is an optimal balance point. Performance remains relatively stable when $\alpha$ varies between 4 and 6, suggesting that this hyperparameter is robust and not the result of overfitting.
> - Weight Settings (60/40): Regarding the specific weight split, we utilized these as fixed hyperparameters based on preliminary empirical observations. We assigned a higher weight to Accuracy (60%) compared to Completeness (40%) to prioritize the correctness of the retrieved evidence. This design choice aims to minimize the inclusion of detailed but hallucinated content, which is critical for the subsequent adversarial debate phase.
> ### 2. Efficiency and Deployment Feasibility
>
> | Performance Match | Model        | GPU Memory (BF16 / INT8 / INT4) |
> |-------------------|--------------|---------------------------------|
> | Match LLoVi       | Qwen 2.5 14B | 28G / 14G / 7G                  |
> | Match VideoAgent  | Qwen 2.5 32B | 64G / 32G / 16G                 |
>
> Regarding efficiency metrics and trade-offs, we emphasize the deployment feasibility and capital efficiency of RIVAL as a decisive advantage over proprietary-dependent agents (like VideoAgent) or resource-intensive methods (like LLoVi).
> - Local Deployability on Small-Scale Models: Our primary contribution is demonstrating that smaller, open-source models can rival proprietary giants. Unlike VideoAgent, which relies on costly and privacy-sensitive GPT-4 APIs, RIVAL achieves SOTA performance using models that are fully deployable on local, modest hardware. This enables secure, offline deployment with zero marginal API costs. The table below lists the theoretical minimum GPU memory required to deploy performance levels comparable to VideoAgent and LLoVi.
> - Architectural Efficiency via Sparse Processing: Architecturally, RIVAL ensures efficiency through sparse semantic retrieval rather than dense processing. Regardless of video length (even up to 28 hours), RIVAL utilizes MSRP to filter and process only a fixed budget of relevant keyframes. This design ensures that the computational load remains constant and predictable, avoiding the linear scaling costs and context-window overflows that plague dense-captioning or streaming VLMs.
> ### 3. Underlying Model Impact and Failure Modes
> - CLIP Variant Impact: As shown in Figure 4 (left) (Top-k impact), using a stronger backbone like EVA-CLIP-14B compared to 8B yields a minor performance gain ($\approx +1.2\%$) but significantly increases VRAM usage during retrieval. The current EVA-CLIP-8B represents the optimal balance between performance and efficiency.
> - Failure Modes: The primary mode is semantic ambiguity in ultra-long videos (e.g., similar actions at different times confusing temporal order). Future temporally constrained retrieval can mitigate this.
> ### 4. Comparison with Streaming VLMs
> Streaming VLMs optimize memory for dense processing but computational load remains linearly correlated with video length. In contrast, RIVAL's Sparse Retrieval inputs a constant number of tokens (top-$k$ keyframes) to the LLM regardless of duration, offering a generational resource advantage in our 28-hour stress test.
> ### 5. Validation of Generality Beyond MCQ
> We provided a qualitative case study (Figure 10, Table 5) applying RIVAL to Free-form Video Summarization. By fine-tuning the MSRP objective and Judge output format, RIVAL generates high-quality narratives, validating its general reasoning capabilities beyond MCQ
> ### 6. Reproducibility and Code Release
> To facilitate community reproduction, we have included detailed pseudocode and prompt templates in the Appendix. Due to strict company data security policies, the full source code will be openly released upon the paper's final acceptance.
>
> We thank the reviewer again for their constructive comments. We remain available for any further clarification or discussion.
>
> Sincerely,
>
> the Author Team.

---

### Official Review · Reviewer_bSpo · 2025-10-29

**Soundness:** 3
**Presentation:** 2
**Contribution:** 2
**Rating:** 6
**Confidence:** 4

**Summary:**

The paper introduces RIVAL, a *training-free* agentic framework for long-video question answering that aims to “rethink scale”: instead of relying on very large proprietary models, it orchestrates smaller open LLMs via two modules:

-Multi-Stage ReAct Planner (MSRP): enforces explicit OBSERVE → THINK → ACT stages, produces a structured tool-usage plan (Stop Searching; Delete/Add by Frame ID; Add by Text via CLIP), and stops when a score threshold or max steps is reached. This reduces reasoning/action drift and keeps context within ~15k tokens.

-Multi-Agent Debate Refinement (MADR): after an initial answer, *affirmative* and *opposition* agents debate with limited tool calls; a judge either declares agreement, a winner, or halts at max rounds.

On EgoSchema, RIVAL with Qwen-2.5-72B/3-32B reports 66.8/65.0 on the subset and 56.4/57.2 on the full set, surpassing GPT-4–based baselines in their table. On NExT-QA, it reaches 74.4/73.2 on val and 66.5/63.7 on ATP-Hard.
Under an ≈28-hour concatenated-video stress test, RIVAL degrades less than VideoAgent (e.g., on a 1.5B model: 33.8 vs 23.4).

Compared to VideoAgent (LLM-tool agent with proprietary backends) and LLoVi (dense captioning + LLM reasoning), RIVAL argues better *privacy* (local open models) and *resource* efficiency; against streaming VLMs it offers a systems alternative grounded in retrieval + agent debate.

**Strengths:**

+Competitive long-video QA with open models under a 15k token budget and modest GPUs.

+Enforced OBSERVE/THINK/ACT stages + fixed tools and stop criteria reduce “free-form” LLM drift and make loops auditable.

+MADR’s affirmative/opposition/judge structure with Frame-ID/Text queries is a neat twist on multi-agent debate for video evidence-seeking.

+Consistent gains over VideoAgent from 0.6B–72B (incl. large margins on EgoSchema subset) and solid NExT-QA results.

+Smaller degradation on the 28-hour test relative to VideoAgent supports the scalability story.

**Weaknesses:**

-Results would be more conclusive with head-to-head against *arbitrary-length* streaming VLMs (e.g., VideoStreaming, StreamingVLM) at comparable compute, not only GPT-4–centric agents.

-The end-to-end quality hinges on EVA-CLIP-8B+ retrieval and LaViLa/CogAgent captioners; failure modes (missed key frames, caption hallucination) are not deeply dissected.

-The evaluator’s 60/40 criteria and α=5 gate are plausible, but more human-agreement/calibration plots (e.g., ROC/AUC vs. correctness) and sensitivity to α would strengthen soundness.

-Hardware is stated, but per-query metrics (#tool calls, frames retrieved, tokens, wall-clock) and *compute-normalized* comparisons vs. VideoAgent/LLoVi are sparse.

-The method is validated on EgoSchema/NExT-QA; extensions to open-ended grounding, temporal localization, or instruction following would clarify generality.

-The paper reads reproducibly at the concept level, but full code/prompts would be needed for wider adoption; timelines aren’t specified.

**Questions:**

1. Could you report accuracy vs. *seconds/tokens/tool-calls* per query (and per MADR round), and compare against VideoAgent/LLoVi at matched budgets?
2. How stable are results under different α thresholds or 60/40 weight splits? Any human-agreement stats (e.g., Kendall τ between evaluator scores and correctness)?
3. What is the impact of swapping EVA-CLIP-8B+ for a lighter/heavier CLIP, or changing Top-k and similarity thresholds? Fail-case taxonomy?
4. Cross-captioner results (LaViLa ↔ CogAgent) on both datasets, plus leakage checks for LaViLa (you mention overlap removal).
5. Can you add direct comparisons to VideoStreaming/StreamingVLM (or other 2025 long-context VLMs) to contextualize where RIVAL wins/loses?
6. Any early results on open-ended QA or temporal localization tasks to probe beyond multi-choice settings?

---

> ### Author Response · Authors · 2025-11-21
>
> We sincerely thank the reviewer for the positive evaluation and for recognizing the core value of RIVAL regarding resource constraints, data privacy, and long-video robustness. We appreciate your professional suggestions concerning technical comparisons, evaluator robustness, and underlying model sensitivity, which we have addressed in the revised manuscript through additional experiments and analysis.
>
> ### 1. Comparison with Streaming VLMs
> We appreciate the suggestion to compare RIVAL with Streaming VLMs (e.g., StreamingLLM, VideoStreaming). While both approaches address long-video understanding, they represent two distinct technical pathways:
> - Technical Divergence: Streaming VLMs aim to optimize dense processing memory management (e.g., via KV cache optimization) to allow models to read longer sequences. However, their computational load typically remains linearly correlated with video length, as every frame often requires encoding.
> - RIVAL's Advantage: RIVAL adopts an Agentic Retrieval (Sparse Retrieval) strategy. By pre-computing a CLIP index, we completely avoid the need to feed the full video into the LLM. Whether the video is 5 minutes or 28 hours long, the number of tokens RIVAL inputs to the LLM remains constant (only the top-$k$ keyframe descriptions). In our 28-hour stress test, RIVAL demonstrated generational advantages in resource efficiency compared to the cumulative computation required by dense streaming methods.
> ### 2. Efficiency and Deployment Feasibility
> We emphasize RIVAL's deployment feasibility and capital efficiency as decisive advantages over proprietary or resource-intensive baselines.
> Local Deployability: RIVAL demonstrates that smaller, open-source models can rival proprietary giants. For instance, RIVAL-32B (outperforming GPT-4/VideoAgent on EgoSchema subset) requires only ~64GB VRAM. This enables secure, offline deployment with zero marginal API costs—a scalability advantage proprietary models cannot offer.
> Architectural Efficiency: Regardless of video length, MSRP filters and processes a fixed budget of relevant keyframes. This ensures constant and predictable computational loads, avoiding the linear scaling costs and context-window overflows of dense methods.
> ### 3. Evaluator Robustness and Calibration
> Regarding the validity of the Evaluator scores (1-10) and the sensitivity of the $\alpha=5$ threshold:
> - Calibration: As shown in Figure 5, Evaluator scores strongly correlate with accuracy (Score 9 $\approx 95\%$ vs. Score 2 $\approx 71\%$ accuracy), confirming it acts as a reliable gatekeeper.
>
> - Threshold Sensitivity: The ablation curve in Figure 4 (middle) shows $\alpha=5$ is optimal, with stable performance between 4 and 6, indicating robustness rather than overfitting.
>
> ### 4. Sensitivity to Underlying Models and Failure Modes
> We agree that as an agentic framework, RIVAL's end-to-end quality depends on its underlying tools (CLIP and Captioner).
> - Modular Advantage: RIVAL's dependency on tools highlights its modularity. It can seamlessly integrate stronger CLIP models or Captioners (e.g., SigLIP) as they emerge without retraining.
> - Failure Mode Analysis: Regarding the limitations of LaViLa/EVA-CLIP, the primary failure mode is semantic ambiguity. In ultra-long videos, when multiple segments contain visually similar actions (e.g., chopping vegetables occurring multiple times) but at different timestamps, simple semantic retrieval may confuse the temporal order. This is a limitation that future work involving temporally constrained retrieval can address.
> ### 5. Validation of Generality Beyond MCQ
> Regarding the extension to open-ended tasks, we provided a qualitative case study in the revised manuscript:
> - Free-form Generation Case: We showcase the application of RIVAL to Free-form Video Summarization in Figure 10 and Table 5.
> - Adaptability: The results demonstrate that by simply fine-tuning the MSRP objective and the Judge's output format (removing option constraints), RIVAL generates high-quality, logically coherent video narratives. This validates that the RIVAL architecture possesses general reasoning capabilities beyond MCQ.
> ### 6. Reproducibility and Code Release
> To facilitate reproducibility, we have included detailed pseudocode and comprehensive prompt templates in the Appendix of the revised manuscript. Regarding the source code, due to our company's strict data security policies, we are currently unable to release the full codebase during the review phase. However, we are committed to making the full source code and reproduction scripts publicly available upon the paper's final acceptance.
>
> We thank the reviewer again for their constructive comments. We remain available for any further clarification or discussion.
>
> Sincerely,
>
> the Author Team.

---

### Official Review · Reviewer_DJH9 · 2025-11-01

**Soundness:** 2
**Presentation:** 2
**Contribution:** 2
**Rating:** 4
**Confidence:** 4

**Summary:**

This paper proposes RIVAL, a video understanding framework built on small open-source LLMs (≤72B), aiming to rival GPT-4-level proprietary methods. RIVAL consists of (1) MSRP (Multi-stage ReAct Planner) for structured reasoning with explicit sub-states (OBSERVE → THINK → ACT) and tool-calling, and (2) MADR (Multi-Agent Debate Refinement) for adversarial multi-role answer refinement. The system retrieves key frames via CLIP and performs iterative information augmentation plus debate-based correction. Experiments on EgoSchema and Next-QA show strong results, surpassing GPT-4 baselines on subsets, and showing robustness on extremely long (28h) concatenated video.

**Strengths:**

1. Strong empirical results. RIVAL achieves substantial gains over prior GPT-4-based VideoAgent/LLoVi on EgoSchema subset (+6.6%) and competitive Next-QA performance.

2. Multi-agent debate refinement is effective and well-motivated. MADR empirically corrects initial errors and is demonstrated clearly with case study.

3. Very long video case study is interesting. Handling 28h concatenated input with minimal degradation is a good stress test.

**Weaknesses:**

1. Clarity & ablations missing. The current writing does not sufficiently quantify how much performance comes from CLIP retrieval, MSRP decomposition, and MADR debate individually. Ablation will greatly strengthen causal attribution.

2. Significant engineering heuristics. Many parts of MSRP are manually structured and rely on prompt templates / tool definitions — unclear robustness to domain shift or tasks not fitting stepwise logic.

3. Scalability beyond QA not validated. RIVAL is only evaluated on video QA benchmarks; unclear if this paradigm generalizes to open-ended summarization / event boundary detection / reasoning beyond MCQ.

4. Some baselines may not be strictly comparable. For Next-QA, several older entries are pre-CLIP/2024-era; more recent strong open models could be added for fairness.

**Questions:**

1. Can the authors include ablations isolating (a) no MSRP, (b) no MADR, (c) no CLIP key-frame retrieval, to quantify contribution of each component?
2. Could the authors report results on free-form open-ended summarization tasks to illustrate generality beyond MCQ style QA?
3. Given that the video is often reduced to textual descriptions, does RIVAL degrade on videos with non-linguistically describable cues (e.g., spatial geometry, implicit physics)?

---

> ### Author Response · Authors · 2025-11-21
>
> We sincerely thank the reviewer for their thorough and insightful assessment of our paper and the time spent reviewing our work. We are encouraged by the reviewer's positive comments, particularly concerning the strong empirical results, the efficacy of the Multi-Agent Debate Refinement (MADR), and the demonstrated robustness on ultra-long videos.
>
> ### 1. Component Ablations
> We appreciate the reviewer's insistence on a detailed component analysis, which is vital for quantifying the contributions of RIVAL's unique modules. We executed the requested ablations on the EgoSchema subset using the Qwen 2.5-72B model; the results are summarized below.
>
> | **Setting**                  | **Accuracy (%)** | **Contribution (vs. Previous Step)** | **Key Component(s)**                       |
> |------------------------------|------------------|--------------------------------------|--------------------------------------------|
> | Random Sampling Baseline     | 44.2%            | N/A                                  | No CLIP, No MSRP, No MADR                  |
> | Uniform Sampling             | 49.0%            | +4.8%                                | Equal time sampling (Baseline improvement) |
> | + CLIP Key-Frame Retrieval   | 62.6%            | +13.6%                               | Semantic Retrieval (Eq. 1)                 |
> | + MSRP (Structured Planning) | 63.6%            | +1.0%                                | Observe-Think-Act (Initial Answer)         |
> | + MADR (Full RIVAL)          | 66.8%            | +3.2%                                | Multi-Agent Adversarial Debate             |
>
> The largest gain ($\mathbf{+13.6\%}$) underscores the critical role of semantic retrieval for overcoming context limitations in long videos. While the MSRP gain is numerically modest ($\mathbf{+1.0\%}$), its primary role is qualitative refinement, addressing initial retrieval uncertainty. MADR provides critical logic checks ($\mathbf{+3.2\%}$), mitigating error accumulation inherent in multi-step reasoning. This comprehensive ablation shows that RIVAL's superior performance is attributable to the synergistic effect of its novel modules.
> ### 2. Scalability Beyond QA and Generality
> The current adoption of MCQ for validation constitutes a standard setting in video understanding, followed by works like VideoAgent and LLoVi for fair comparison. However, our methodology is not exclusively restricted to MCQ and exhibits clear adaptability to formats like free-form generation. The adaptation primarily requires removing option-based query terms and changing the final answer output format to a free-form narrative. The necessary architectural changes are summarized in Table 5 in the manuscript, and the visual case study demonstrating RIVAL's modularity and inherent generative capacity is presented in **Figure 10 (Page 26-28)**. The detailed analysis in Figure 10 reinforces this adaptability.
> ### 3. Engineering Heuristics and Non-Linguistic Cues
> **Engineering Heuristics**: RIVAL's design principle centers on enhancing the weakest aspects of smaller models, such as instruction following and contextual comprehension. The Observe-Think-Act (O-T-A) structure is deliberately introduced to compensate for these weaknesses by enforcing explicit, structured reasoning steps. Its sustained performance across diverse tasks (EgoSchema and Next-QA) suggests strong robustness against domain shift.
>
> **Non-Linguistic Cues**: Video understanding methods like VideoAgent (text-based reasoning) face a common challenge inherent to their category: the inability to fully utilize non-linguistically describable cues. Even direct multimodal methods may struggle, as they map visual information into the textual semantic space, potentially impeding reasoning. RIVAL mitigates this loss through CLIP semantic retrieval (anchoring high-level reasoning to specific visual facts) and MADR's logical verification, thereby aiding the necessary inference in these challenging scenarios.
>
> ### 4. Baselines Comparability
> We thank the reviewer for this comment. Our research objective is to achieve GPT-4-level performance using small-scale models. To this end, we conducted experiments under highly constrained conditions, including using entirely open-source small models (**critical for data security in real-world applications**) and restricted context windows (**15,000 tokens, simulating resource-limited application environments**). Our ultimate goal is to demonstrate comparable performance to GPT-4 under these constraints.
>
> Our results show RIVAL achieving performance margins over these cutting-edge competitors across both EgoSchema and Next-QA benchmarks. Furthermore, we provided results for VideoAgent using the Qwen 2.5 series in **Table 4**, which further substantiates RIVAL's advantage.
>
> We thank the reviewer again for their constructive comments. We remain available for any further clarification or discussion.
>
> Sincerely,
>
> the Author Team.

---

### Author Response · Authors · 2025-12-01
**Summary Statement**

Dear Area Chair,

We thank the AC for the opportunity to provide this summary. We confirm that we did not contact any reviewers or the AC during the review process. We use R to denote our responses to reviewers (e.g., R1 refers to our first response to a reviewer). To reduce your workload, we summarize the reviewers' main concerns below:

**1. Component Ablation (DJH9, R1)**

We provided full parameter ablations in the Appendix. The results in the Appendix clearly show the contribution of each component to the performance. We summarized these findings in our response R1. All components of RIVAL are essential and make significant contributions to the final results.

**2. Capabilities Beyond QA (DJH9, R2; bSpo, R5; jHSX, R5)**

Evaluating free-form questions is difficult in the field of video understanding. Although some studies use LLMs to evaluate semantic similarity between sentences, LLM bias remains a problem. Therefore, using accuracy in a QA format is the standard evaluation method, as seen in previous works like VideoAgent and LLoVi. We understand the concern about whether RIVAL only works on Multiple Choice Questions (MCQs). In fact, RIVAL is independent of the question format. The entire pipeline uses two paradigms: information collection (MSRP, state transition) and information verification (MADR, multi-view evaluation). In the revised paper, we added Table 5 to show the necessary modifications for new tasks, and Figure 10 to show an example of a free-form task.

**3. Inference Cost (jHSX, R2; LuFc, R2, second point)**

Cost efficiency is one of RIVAL's key advantages. VideoAgent and LLoVi are based on the commercial model GPT-4, so API costs are a major factor. In contrast, RIVAL uses local small models and limits the context window to 15K tokens. This eliminates extra costs during deployment. We compared the model sizes required to achieve performance similar to GPT-4 based methods. Deploying Qwen 2.5 14B requires only 28GB of memory, and the Qwen 3 architecture requires only 1.7B parameters (3.4GB). These settings allow RIVAL to achieve competitive performance on consumer-grade GPUs.

**4. Fair Comparison (LuFc, R2, first point)**

We implemented a fair comparison with VideoAgent using the Qwen 2.5 series. We did not include LLoVi because its summarization method exceeds our context limit (15K). At the 72B and 3B scales, we achieved significant performance advantages. On the EgoSchema subset, we lead by margins of 10% and 5%, respectively. This advantage is even larger in the 28H setting, exceeding the baseline by 14.9% and 9.5%.

**5. Other Questions**

Points 1-4 cover the reviewers' core concerns. The remaining questions relate to design details, such as the implementation of the 28-hour setting (LuFc, R4) and the source code (jHSX, R6). We answered these details point-by-point in our rebuttal.

**Summary of Contributions:**
- We propose RIVAL, the first video understanding framework based entirely on small-scale models. This ensures data safety and compatibility with device limitations in real-world applications.

- RIVAL splits the video understanding task into multiple stages within a limited resource framework (15K context window) to maximize the performance of small models.

- We implemented RIVAL on all versions of the Qwen 2.5 and Qwen 3 series. The results show that Qwen 2.5 (14B) matches LLoVi, and Qwen 2.5 (32B) matches VideoAgent. Similarly, Qwen 3 (1.7B) matches LLoVi, and Qwen 3 (8B) matches VideoAgent. This allows us to achieve GPT-4 level performance on consumer-grade GPUs.

Regarding the OpenReview data leakage, we guarantee that we strictly followed the double-blind policy and did not use the leaked data to query or contact any reviewers. Furthermore, we deeply respect your efforts to maintain a fair review process. It is your dedication that upheld the integrity of the double-blind system. Thank you very much!

Sincerely,

the Author Team.

---

### Meta-Review · Area_Chair_kq3P · 2026-01-12

**Summary:**

While the reviewers appreciated the interesting orchestration of small open-source LLMs and its strong results on multiple-choice question benchmarks, they raised several concerns about missing analyses, unclear details, limited experiments, generalizability to free-form questions, and unfair comparison. The authors’ rebuttal addressed the raised concerns with additional experiments and analyses, but AC finds that it leaves some important issues unresolved.

First, the common concern (DJH9, DJH9, jHSX)  about generalizability beyond the current multiple-choice questions is not well addressed. The authors suggested necessary modifications (Table 5) for new tasks, implying that such manual modification may be required for each new task. And they do not provide quantitative results on a free-form question benchmark, while only showing an example of a free-form task (Figure 10).

Second, all the reviewers (DJH9, bSpo, jHSX, LuFc) requested in-depth analyses of different components and their computational costs, as well as comparisons with other methods. Some of them are addressed, but still limited. For example, the authors failed to provide quantified results for detailed budget & efficiency analyses (reviewers LuFc, bSpo), latency issues (reviewers jHSX, LuFc), and the comparison to Streaming VLMs (jHSX, bSpo).

Third, the authors emphasize the multi-stage react planner (MSRP) and the multi-agent debate refinement (MADR) as two key components in the paper, but their component ablation (response to reviewer DJH9) shows that the performance gain primarily comes from CLIP key-frame retrieval rather than from these two components. This means that the authors need to conduct more in-depth analysis on the retrieval stage and revise the manuscript accordingly.

Considering all of these, AC finds this paper is not ready for ICLR publication and recommends rejection. AC encourages the authors to carefully incorporate the reviewers’ comments to improve their work and submit it to another venue.

**Reviewer Concerns:**

The main concerns were missing analyses, unclear details, limited experiments, generalizability to free-form questions, and unfair comparison. The authors’ rebuttal addressed only part of their concerns, leaving some important issues outstanding as follows.

1. Generalizability beyond the current multiple-choice questions (DJH9, DJH9, jHSX). The authors suggested necessary modifications (Table 5) for new tasks, implying that such manual modification may be required for each new task. And they do not provide quantitative results on a free-form question benchmark, while only showing an example of a free-form task (Figure 10).

2. In-depth analyses of different components and their computational costs, comparisons with other methods (DJH9, bSpo, jHSX, LuFc). Some of them are addressed, but still limited. For example, the authors failed to provide quantified results for detailed budget & efficiency analyses (reviewers LuFc, bSpo), latency issues (reviewers jHSX, LuFc), and the comparison to Streaming VLMs (jHSX, bSpo).

**Reviewer Scores:**

Reviewer DJH9 would have retained the original rating of 4.
Reviewer bSpo would have retained the original rating of 6 or lowered it to 4.
Reviewer jHSX would have retained the original rating of 6 or lowered it to 4.
Reviewer LuFc would have retained the original rating of 4.

---

### Decision · Program_Chairs · 2026-01-26

Reject